

# Coupling a large-scale glacier and hydrological model (OGGM v1.5.3 and CWatM V1.08) – Towards an improved representation of mountain water resources in global assessments

Sarah Hanus[1], Lilian Schuster[2], Peter Burek[3], Fabien Maussion[2,4], Yoshihide Wada[5], and Daniel Viviroli[1]

[1]Department of Geography, University of Zurich, Zurich, Switzerland
[2]Department of Atmospheric and Cryospheric Sciences (ACINN), Universität Innsbruck, Innsbruck, Austria
[3]International Institute for Applied Systems Analysis, Laxenburg, Austria
[4]Bristol Glaciology Centre, School of Geographical Sciences, University of Bristol, Bristol, United Kingdom
[5]Climate and Livability Initiative, Center for Desert Agriculture, Biological and Environmental Science and Engineering Division, King Abdullah University of Science and Technology, Thuwal, Saudi Arabia

**Correspondence:** Sarah Hanus (sarah.hanus@geo.uzh.ch)

**Abstract.** Glaciers are present in many large river basins, and due to climate change, they are undergoing considerable changes in terms of area, volume, runoff and seasonality. Although the spatial extent of glaciers is very limited in most large river basins, their role in hydrology can be substantial because glaciers store large amounts of water at varying time scales. Large-scale hydrological models are an important tool to assess climate change impacts on water resources in large river basins worldwide.

Nevertheless, glaciers remain poorly represented in large-scale hydrological models. Here we present a coupling between the large-scale glacier model OGGM (v1.5.3) and the large-scale hydrological model CWatM (V1.08). We evaluated the improved glacier representation in the coupled model against the baseline hydrological model for selected river basins at 5 arcmin resolution and globally at 30 arcmin resolution, focusing on discharge projections under climate change scenarios. We find that increases in future discharge are attenuated, whereas decreases are exacerbated when glaciers are represented explicitly in

the large-scale hydrological model simulations. This is explained by a projected decrease in glacier runoff in almost all basins. Calibration can compensate for lacking glacier representation in large-scale hydrological models in the past. Nevertheless, only an improved glacier representation can prevent underestimating future discharge changes, even far downstream at the outlets of large glacierized river basins. Therefore, incorporating a glacier representation into large-scale hydrological models is important for climate change impact studies, particularly when focusing on summer months or extreme years.

## 1  Introduction

Mountains are frequently referred to as "Water Towers" because of their disproportionally high runoff and the delay in water release caused by the mountain cryosphere (Viviroli et al., 2007; Immerzeel et al., 2020; Viviroli et al., 2020). Snow accumulation during winter and snow melt during spring and summer are essential components of the seasonal redistribution of water (Barnett et al., 2005; Mankin et al., 2015). Glaciers, on the other hand, act as storage both seasonally and over the long term



(Kaser et al., 2010). Their seasonal melt pattern contributes to discharge during the summer and late summer months (van Tiel et al., 2021).

    Due to climate change, glaciers are experiencing considerable mass loss (Marzeion et al., 2020; Rounce et al., 2023). As glaciers lose mass, their annual and seasonal melt volumes increase until a maximum, called "glacier peak water", is reached. Thereafter, melt volumes decline due to continued glacier volume decrease or stabilization, leading to lower seasonal and

annual melt volumes compared to the years around glacier peak water. If the climate stabilizes, glaciers will eventually reach a smaller stable state, with lower melt volumes, which leads to lower seasonal but similar annual runoff volumes compared to their previous larger stable state condition (Huss and Hock, 2018). Globally, glacier peak water has already been surpassed in around half of the glacierized river basins worldwide (Huss and Hock, 2018).

    A recent study concludes that the contribution of snowmelt is more important for streamflow than glacier melt contributions

in Asia (Kraaijenbrink et al., 2021). However, Lutz et al. (2014) show the importance of glaciers for streamflow in the upper basins of the Himalayas. Furthermore, the global study by Huss and Hock (2018) suggests that glaciers contribute essentially to streamflow in major river basins and that streamflow will decrease by more than 10% during at least one month of the melting season in one-third of the world's glacierized basins due to decreased water contribution by glaciers. Even in regions experiencing declining streamflow over the 21$^{st}$ century, glacial runoff has the potential to buffer droughts (Ultee et al., 2022).

The changing melt seasonality and annual contributions of glaciers to streamflow make it essential to consider glacial melt in hydrological models to simulate future hydrological changes in partially glacier-covered basins realistically and to assess water availability. Moreover, incorporating glaciers into hydrological modelling allows us to better understand and quantify the importance of glaciers compared to other runoff components in the past and future. This can provide valuable insights into the importance of glaciers for water resources.

In catchment modelling, hydrological models have included glacier routines (see van Tiel et al. (2020) for an overview). To model glaciers in a changing climate realistically, glacier area and volume changes should be considered. For example, Seibert et al. (2018) introduce a glacier routine for the semi-distributed model HBVlight that connects mass balance changes to changing glacier area. Some models use volume-area scaling (Lutz et al., 2014) or specific ice-flow models (Wijngaard et al., 2018; Biemans et al., 2019) to represent changing glaciers. In contrast, other studies derive changes in glacier extent from

outputs of glacier modelling (e.g. Muelchi et al., 2021; Hanus et al., 2021). These methods are often applied to catchments smaller than 5,000 km$^2$ and only rarely to catchments larger than 150,000 km$^2$ (van Tiel et al., 2020). Most of these hydrological models do not include human water use. However, in many river basins human water use cannot be neglected. Therefore, some regional studies have simulated hydrological processes upstream with a glacio-hydrological model and downstream with a model which incorporates human water use (e.g. Wijngaard et al., 2018; Biemans et al., 2019).

For regional to global studies on water resources, so-called global hydrological models are frequently used to assess water availability and human water use (Schewe et al., 2014). Throughout this study, we refer to these models as large-scale hydrological models because they are increasingly used at a regional scale in addition to a global scale. Large-scale hydrological models mostly lack a glacier representation. Only one out of 16 models used to simulate water availability in the Inter-Sectoral Impact Model Intercomparison Project phase 2 (ISIMIP2) had some representation of mountain glaciers, namely CWatM (Tel-



teu et al., 2021). However, even this model has a very simplistic glacier representation resembling a snow redistribution method
to avoid snow accumulation (Burek et al., 2020; Telteu et al., 2021).

Efforts were made to improve the glacier representation in large-scale hydrological models. An energy balance ice melt
method was incorporated in VIC for a regional analysis in China by Zhao et al. (2013). This approach was further developed
by Su et al. (2016) to assess future streamflow in various Asian basins. The authors applied volume-area scaling to derive initial
glacier volume based on glacier area at the basin-scale and adapted the glacier volume and area accordingly every 10 years. To
our knowledge, two attempts exist to couple a global glacier model with a large-scale hydrological model to improve glacier
representation. In one study, glacier model outputs were incorporated into the WaterGAP model to assess past water storage
changes (Cáceres et al., 2020). In a second study, glacier model outputs were incorporated into the PCR-GLOBWB model to
assess potential improvements in model performance (Wiersma et al., 2022). Both studies focused on past simulations and did
not evaluate the effect of glacier representation in large-scale hydrological models on future discharge projections, although
glacier runoff is expected to change substantially during this century. Moreover, the model coupling has not been described in
detail nor made publicly available.

This confirms that glacier melt remains poorly represented in large-scale hydrological models to date. Therefore, large-scale
hydrological models lack the representation of this rapidly changing water source. This is problematic because these models
are used as tools for climate change impact assessments. Their primary objective is to answer questions about future water
availability, water use and its spatial and temporal patterns. Hence, our primary objective was to improve the representation
of glaciers, and consequently an important aspect of mountain water towers, within a large-scale hydrological model. This
enhancement aims to capture the effect of glacier melt and retreat on water availability.

Coupling a state-of-the-art glacier and hydrological model has the potential to improve the glacier representation by lever-
aging the most recent advancements from the glacier modelling community. We think it is beneficial to use a tested and
well-established glacier model as opposed to developing a new glacier routine. Additionally, an analysis of the challenges
when coupling such models can provide insights into how the hydrological and glacier modelling communities can foster con-
tinued collaboration in the future.

Hence, this paper introduces a framework to couple the Open Global Glacier Model v1.5.3 (OGGM; Maussion et al., 2019)
and the Community Water Model V1.08 (CWatM; Burek et al., 2020) on 5 arcmin and 30 arcmin resolution. Both state-of-the-
art models are openly available. This framework facilitates an explicit inclusion of glacier runoff in large-scale hydrological
modelling through dynamic modelling of glaciers. In the methods, the coupling framework is introduced in detail. First, the
framework is evaluated using selected major river basins in Europe and North America on 5 arcmin resolution. Second, a
similar evaluation is conducted globally at 30 arcmin. Future changes in mean monthly and mean annual runoff are com-
pared between the coupled model and the original model to assess the influence of the coupling on climate change impact
assessments.





## 2 Methods: model coupling

### 2.1 Hydrological model

The large-scale hydrological model Community Water Model (CWatM) is an open-source model developed by the International
Institute for Applied System Analysis (IIASA) (Burek et al., 2020). The model includes all dominant hydrological storages
and processes, such as snow, soil, groundwater and river routing. Additionally, it can simulate human water use and reservoir
regulations. For an in-depth model description, we refer to Burek et al. (2020). CWatM is used at 5 arcmin and 30 arcmin
resolution, but can also be used at 1 km resolution (Guillaumot et al., 2022). At 30 arcmin resolution, CWatM was used in
the Inter-Sectoral Impact Model Intercomparison Project (ISIMIP) rounds 2b and 3, which compares large-scale hydrological
model outputs (Frieler et al., 2017; Warszawski et al., 2014). CWatM has extensive and publicly available documentation
of the source code, the model structure and model training and tutorials (https://cwatm.iiasa.ac.at/). Together with its modular
structure, this makes CWatM a highly suitable candidate for the model coupling. CWatM has a simple glacier representation: At
high-elevation zones where snow has not melted until the beginning of summer, the melting rate is increased and the average
temperature of the grid cell is applied. This implicitly simulates the downward movement of glaciers (Burek et al., 2020).
However, this glacier representation has several shortcomings. First, it assumes that glaciers are in equilibrium, meaning that
only snow previously accumulated is melted. This representation neglects the change in glacier runoff when glaciers retreat, a
phenomenon that is ongoing (Hugonnet et al., 2021) and projected to continue throughout this century (Rounce et al., 2023).
Second, it assumes the presence of glaciers wherever snow accumulates in the model. We argue that this glacier representation
is more akin to a snow redistribution function used to avoid excessive snow accumulation (Freudiger et al., 2017; Burek et al.,
105 2020).

### 2.2 Glacier model

The Open Global Glacier Model (OGGM) framework is a community-driven, well-documented and open-source global glacier
model (Maussion et al., 2019). This makes it stand out from a number of global glacier models which are used for global glacier
mass change projections (Marzeion et al., 2020). OGGM has been used in global studies (Marzeion et al., 2020; Rounce et al.,
2023) but also on regional and basin scales (e.g. Tang et al., 2023; Furian et al., 2022; Yang et al., 2022). OGGM has a modular
structure, which is beneficial for new developments and a variety of research questions requiring different model setups.

Each glacier is simulated individually by OGGM. For this study, OGGM used the glacier outlines from the Randolph Glacier
Inventory (RGIv6.0, Pfeffer et al., 2014) and elevation-band flowlines (e.g., Huss and Farinotti, 2012; Werder et al., 2020)
derived from a digital elevation model. A one-dimensional shallow-ice flowline model of OGGM approximates the influence
of ice flow on glacier dynamics. A temperature index mass-balance model computing mass-balances at each elevation band
was calibrated for every glacier by glacier-wide geodetic observations (2000–2019, Hugonnet et al., 2021). The ice volume
consensus estimates of Farinotti et al. (2019) at the RGI year (often near the year 2000) have been matched regionally by
calibrating the ice creep parameter $A$.



For a detailed model description, we refer to Maussion et al. (2019) and the online documentation (https://docs.oggm.org/).
The model framework is under continuous development and now possesses the capability to simulate at a daily resolution
using a daily mass balance model and daily climate data, a novel development for large-scale glacier models (Schuster et al.,
2023). In the context of hydrological applications, it is relevant to mention that the utilized OGGM version (v1.5.3) does not
differentiate between snowmelt and ice melt on glaciers.

### 2.3   Model differences in the representation of elevation, precipitation and snowmelt

Both models were slightly adapted to facilitate coupling: In CWatM, we incorporated the rain-snow partitioning function from
OGGM. OGGM was modified to provide daily outputs for melt and rainfall on glaciers. Nevertheless, differences between the
two models in the representation of snow accumulation and melt processes remain (Table 1).

An important difference is the spatial resolution of snow accumulation and melt processes. CWatM has a maximum of ten
evenly large elevation zones per grid cell (at 5 arcmin resolution, one elevation zone per ~10 km$^2$) for which snow processes
are modelled individually. In OGGM, the snow and ice processes are modelled on a much finer scale. Each glacier is split into
elevation zones of 30 m height difference. Also, the spatial resolution of the model output is different, as OGGM outputs are
given per glacier, whereas CWatM inputs and outputs are given per simulation grid cell.

Another relevant difference between the models is the correction of precipitation input data. Precipitation amounts are often
underestimated in mountain areas, but the magnitude of this error remains difficult to assess (Beck et al., 2020; Azam et al.,
2021). Glacier models use a precipitation factor to correct for measurement errors and additional errors in snow input, such as
those caused by avalanches, thereby allowing for better alignment with mass balance data (Hock et al., 2019; Rounce et al.,
2020). In CWatM, a multiplicative factor for snowfall correction is implemented as an optional calibration parameter following
the rationale that snowfall has a larger error range than rainfall (Beck et al., 2020). Following the same rationale, we applied
the precipitation factor in OGGM only to snow and not to rain. However, efforts to further harmonize the two models by using
the same precipitation correction factor resulted in unrealistic calibration and a deterioration in model performance. Therefore,
we decided to retain the snow correction procedures of each model, although this leads to disparities between the two models.
Thus, the precipitation factor in OGGM was calibrated to a fixed value across all glaciers in the studied region to best represent
the observed inter-annual mass-balance variability from WGMS (2020), which is the standard procedure in OGGM v1.5.3. In
CWatM, the snowfall correction factor was only calibrated when there was clear evidence of precipitation underestimation in
the river basin. In other cases, no snowfall correction was applied.

Both models use a temperature-index model to represent snow and ice melt, with CWatM including melt seasonality and the
effect of increasing melt during rainfall. Temperature-index models which use a degree-day factor, i.e. snowmelt coefficient,
are widely used in hydrological models to calculate snow melt. They have low data requirements which is beneficial for large-
scale studies (Girons Lopez et al., 2020). The utilized OGGM version (v1.5.3) uses one combined degree-day factor for snow
and ice melt. This parameter was calibrated per glacier using 20-year average geodetic mass balance data from Hugonnet et al.
(2021). However, in CwatM the degree-day factor was calibrated jointly with other model parameters using discharge data
(Table 1), resulting in one parameter value for the whole basin area upstream of the gauge used for calibration.





**Table 1.** Overview of model inputs, resolution and outputs of the applied CWatM and OGGM versions and their configurations

| | CWatM | OGGM |
|---|---|---|
| Model version | 1.08 | 1.5.3 + OGGM daily massbalance model |
| Time-step | Daily | Daily |
| Vertical resolution of snow calculations | Up to 10 equally sized elevation zones per grid cell. Elevation zones have the same area but different elevation range | Elevation zones with a range of 30m on each glacier. Elevation zones have the same elevation range but different areas |
| Correction of meteo data to elevation zones | Temperature lapse rate: -0.0065 K km$^{-1}$, No precipitation lapse rate | Temperature lapse rate: -0.0065 K km$^{-1}$, No precipitation lapse rate |
| Output resolution | Per grid cell (30', 5' arcmin ($\sim$50, 10 km)) | Per glacier |
| Input data | Meteo data (gridded), other gridded input maps, e.g. land use, soil and aquifer properties, drainage direction (cf. Burek et al., 2020) | Meteo data (gridded), glacier outlines from the Randolph Glacier Inventory (RGIv6.0, Pfeffer et al., 2014), a digital elevation model |
| Meteo variables | precipitation and temperature; for evaporation calculation: min/max temperature, surface pressure, wind speed, solar radiation, humidity | precipitation and temperature |
| Precipitation correction | optional snow correction with a multiplicative factor | Basin-specific or global precipitation correction factor |
| Melt representation | Degree-day method including seasonality and the effect of increasing melt during rainfall, melt threshold at 0°C | Degree-day method, melt threshold at 0°C |
| Partitioning of rain and snow | two thresholds (default: 0 and 2°C); linear snow and rain proportion change in between | two thresholds (default: 0 and 2°C); linear snow and rain proportion change in between |
| Calibration parameters | Selectable, with a list of 12 suggested parameters (https://cwatm.iiasa.ac.at/calibration.html) including the degree-day factor which governs snow melt | glacier-dependent degree-day factor and regional creep parameter, precipitation correction factor, temperature bias |
| Operating system | Windows, Linux, MacOS | Linux, MacOS |
| Can be run on a personal laptop? | Yes, for single basins at 5 arcmin resolution or world at 30 arcmin resolution | For selected basins with less than a few hundred glaciers |

## 2.4 Model coupling approach

The main rationale behind the coupling was to adhere to the modular philosophy of the models and maintain an open framework
to enable their continuous development. Therefore, it was decided to implement a one-way sequential coupling where first
OGGM was run and the output was translated via a pipeline, consisting of a set of functions, to input for CWatM (Fig. 1).
Like this, the glacier output could potentially also be used with other hydrological models that use NetCDF files as input files.





OGGM handled the glacierized portions of the modelling domain, while CWatM handled the non-glacierized portions. The glacier area, which was updated each year, determined the extent of the model domain addressed by OGGM and therefore excluded from CWatM simulations. In the modelling chain, the following sequence was followed: first, the glaciers within the modelled river basin were identified. Second, OGGM was calibrated and run for these glaciers. Third, the pipeline translated output from OGGM into a format readable as input for CWatM. Finally, CWatM was run with glacier area and melt as input data. The coupling is explained in more detail in the following sections.

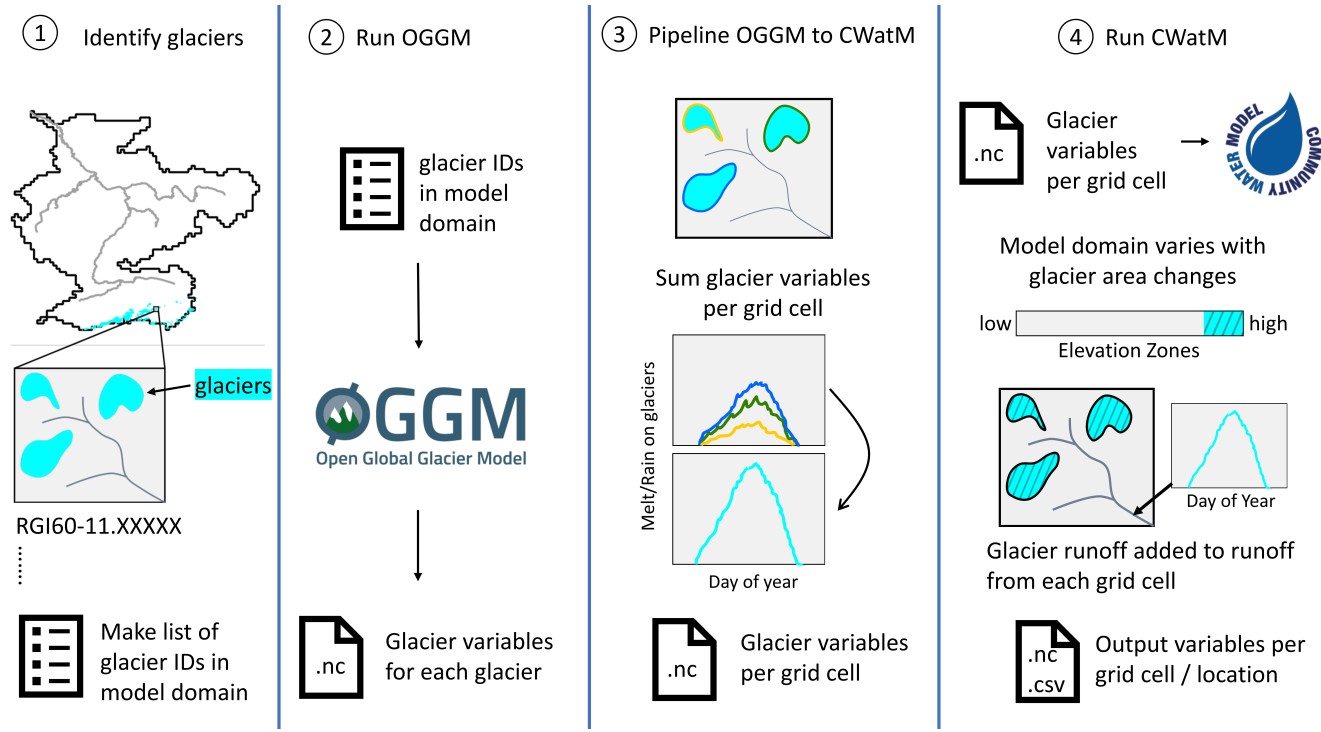

**Figure 1.** Schematization of the steps to sequentially couple the glacier model (OGGM) and the hydrological model (CWatM)

### 2.4.1 Run OGGM for selected glaciers

To model the glaciers in the model domain with OGGM, the glaciers had to be identified using RGIv6.0 (Fig. 1 step 1). The IDs of these glaciers were stored as a list which was used by OGGM. OGGM was calibrated and run dynamically, on a daily timescale. The relevant output variables of OGGM for the hydrological model are the glacier area ("on_area"), the melt ("melt_on_glacier") and the rainfall amount on the glacier ("liq_prcp_on_glacier"). These variables for each modelled glacier were jointly stored in a NetCDF file (Fig. 1 step 2). Note that we ignore the off-glacier output variables from OGGM, as we model this domain by CWatM.





**Translate OGGM output to CWatM input**

CWatM requires input data per grid cell in NetCDF format (latitude, longitude, time dimensions). Therefore, OGGM output per glacier had to be translated to CWatM input per grid cell (Fig. 1 step 3). The following steps were used to generate raster NetCDF files of yearly glacier area and daily glacier melt and rain on glaciers.

One glacier can cover several grid cells and one grid cell can contain several glaciers. Therefore, the geometric glacier outlines of RGIv6.0 were intersected with the model grid at WGS84 and then reprojected to Eckert IV to calculate the glacier area per grid cell at RGI date similarly to Li et al. (2021). To derive the glacier area of each glacier $g$ at time $t$ in grid cell $i$ ($A_{g,t,i}$), the relative change in glacier area was multiplied by the glacier area contained in the grid cell (Eq. 1). Afterwards, the total glacier area within one grid cell was calculated for every year (Eq. 2). In case it exceeded the area of the grid cell due to

glacier growth, the surplus area was redistributed evenly to the four neighboring grid cells.

    In reality, the glacier area decreases strongest close to the terminus of the glacier, when the glacier melts, whereas during glacier growth, the volume at the top of the glacier increases and subsequently the area at the terminus. However, the coupling was designed for large-scale modelling, and therefore, these small-scale dynamics are neglected. We assume the same relative area reduction or growth for all grid cells that a glacier covers (Fig. S1.1).

$$A_{g,t,i} = \left(\frac{A_{g,t}}{A_{g,RGI}}\right) \cdot A_{g,RGI,i} \tag{1}$$

$$A_{t,i} = \sum_{g=1}^{G} A_{g,t,i} \tag{2}$$

    The rain and melt amount over a glacier were routed to the grid cell in which the terminus of the glacier is located at RGI date. The glacier melt includes snow and ice melt on the glacier area. The glacier melt and rain on all glaciers with terminus location within the same grid cell were summed to obtain the total glacier melt and rain on glacier per grid cell (Eq. 3, 4).

$$M_{t,i} = \sum_{g=1}^{G} M_{g,t,i} \tag{3}$$

$$R_{t,i} = \sum_{g=1}^{G} R_{g,t,i} \tag{4}$$

### 2.4.2   Implementing glacier runoff in CWatM

CWatM uses up to ten elevation zones per grid cell to simulate snow processes more realistically. The elevation difference

within one pixel is assumed to be normally distributed. To exclude the glacier area within each grid cell from CWatM simulations, the glacier area was subtracted from the grid cell area starting from the highest elevation zone of each grid cell. The



underlying assumption was that glaciers are located at the highest elevations within each grid cell (Fig. 1 step 4, S1.1). To include the glacier melt in CWatM, an option was implemented to add the glacier melt and rain on glaciers to the surface runoff per grid cell, under the assumption that rain on glaciers and glacier melt cannot infiltrate into the soil or groundwater.

The model setup without glacier coupling is called the baseline model (CWatM$_{base}$) and the model setup including glaciers is referred to as the coupled model (CWatM$_{glacier}$). Moreover, an additional mode for running CWatM has been implemented to quantify the glacier melt contribution to discharge. This involves simulations that exclude glacier areas and runoff contributions from glaciers (CWatM$_{glacier,bare}$).

## 3 Methods: model coupling evaluation

The model coupling was evaluated on two different use cases: first, for individual well-monitored large river basins with a specific calibration at 5 arcmin resolution; and second, for global simulations at 30 arcmin resolution. The rationale behind employing two different resolutions was to assess the suitability of the coupling for assessments in large river basins and global assessments. For the former, a 5 arcmin resolution is often used, whereas global assessments such as ISIMIP often use 30 arcmin resolution.

The first use case gives detailed insights into the effect of the coupling on discharge simulations by comparing CWatM$_{base}$ and CWatM$_{glacier}$ simulations to observations, and reveals benefits and limitations at the river-basin scale. The second case shows how coupling performs at global scale without the explicit calibration of individual river basins, and highlights the challenges and differences compared to modeling at the basin-scale. This is relevant for global climate impact assessments of water availability which are often based on large-scale hydrological models.

As daily meteorological input data for OGGM and CWatM we used the GSWP3-W5E5 dataset (Cucchi et al., 2020), which is also employed in ISIMIP3a simulations. For the evaluation at 5 arcmin, we downscaled GSWP3-W5E5 with WorldClim v2.1 (Fick and Hijmans, 2017) using bilinear interpolation. For future simulations, the meteorological forcing data of five GCMs bias-corrected with the GSWP3-W5E5 data set were used (GFDL-ESM4, UKESM1-0-LL, MPI-ESM1-2-HR, IPSL-CM6A-LR, MRI-ESM2-0), each forced with a low and high emission scenario (Shared Socioeconomic Pathways: SSP1-2.6,

SSP5-8.5). These were accordingly downscaled to 5 arcmin resolution. The median global temperature increase projected by the GCMs for 2070–2099, in comparison to the pre-industrial level, was +1.9°C and +4.2°C, respectively.

CWatM requires additional input data, e.g. flow direction map, information on the land cover type, water demand, soil and groundwater characteristics, for which the global maps described in Burek et al. (2020) were used.

### 3.1 Calibration of selected river basins (5 arcmin)

To evaluate the model coupling on a river basin scale, four large river basins with glacial influence in Europe (Rhine, Rhone, Glomma) and North America (Fraser) were used (Fig. 2). These river basins are well-monitored and have long-term observed discharge time series available at an upstream and downstream gauge. The percentage of glacierized area is highest for the



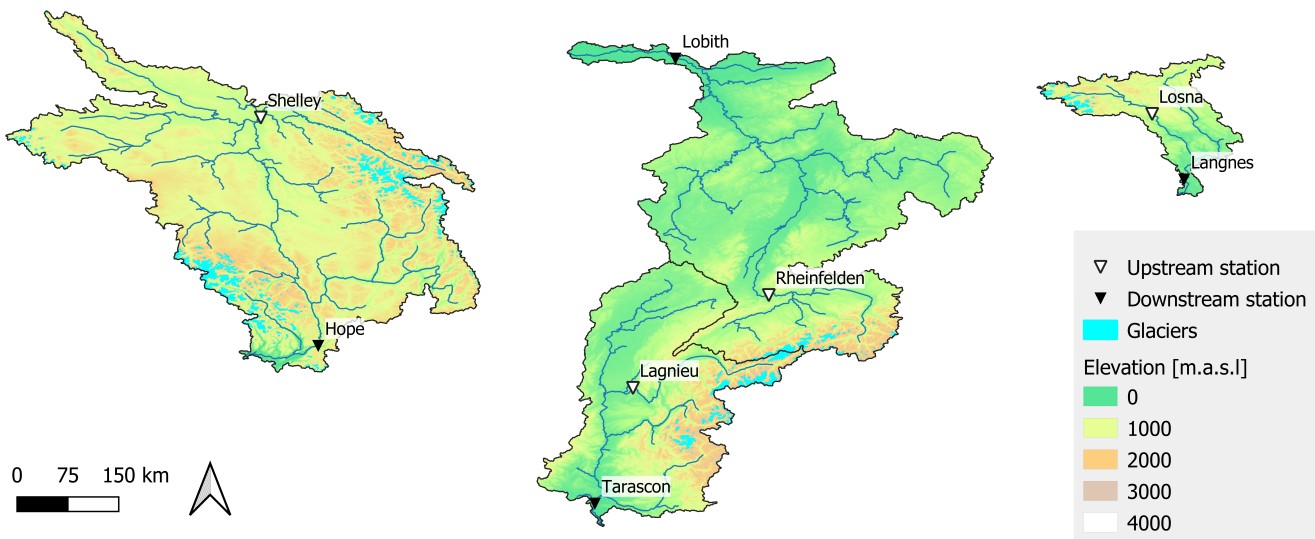

**Figure 2.** Glacierized study basins selected for model coupling evaluation at 5 arcmin with explicit calibration. The triangles show the location of discharge gauges used for calibration. Elevations shown are based on DEM by Yamazaki et al. (2019).

Fraser (1.2%), followed by the Rhone (1%), Glomma (0.7%), and Rhine (0.2%) for the region upstream of the downstream gauge.

A genetic algorithm (NSGA-II from python package DEAP, Fortin et al. (2012)) was used to calibrate 12 parameters in CWatM that govern major hydrological fluxes (Burek et al., 2020). The parameters were calibrated separately for the upstream and downstream regions of each basin using two gauges (see Fig. 2). Discharge data were obtained from GRDC (GRDC, 2022) and HydroPortail (HydroPortail, 2022). As objective function, the non-parametric version of the Kling Gupta Efficiency (KGE) was used (NPE, Pool et al., 2018) (weight 0.8) in combination with a penalty for snow cover error to reduce excessive snow
accumulation in summer (weight 0.2).

CWatM$_{\text{base}}$ and CWatM$_{\text{glacier}}$ were both calibrated for each study basin for the time period 2000–2009 to evaluate the effect of explicit glacier inclusion in the model. Each setup was calibrated five times to capture parameter uncertainty. The performance of the five calibrated parameter sets per river basin was evaluated for the time period prior to calibration (1990–1999).

### 3.2 Global parameter sets (30 arcmin)

To evaluate the model coupling on a global scale, global parameter sets were used. Most large-scale hydrological models employ one single parameter set for the entire globe and do not undergo an explicit calibration (Telteu et al., 2021). Here we used the parameter set, which was used in ISIMIP 3 simulations of CWatM. To consider parameter uncertainty we obtained additional five parameter sets by calibration of CWatM$_{\text{base}}$. We used 601 GRDC stations which complied with criteria laid out in Burek and Smilovic (2022) and had at least 5 years of discharge data in the calibration period. We used the period



from 1985 to 1994 for calibration to maximize the number of discharge stations available and calibrated the 12 parameters to achieve the best possible performance over all stations similar to Greve et al. (2023). The performance was evaluated for the period 2004–2013 to maximise the discharge observation length available per river basin. To limit runoff differences between the model setups to the effect of glacier inclusion, the global calibration procedure was not repeated for CWatM$_{glacier}$ and the parameters of CWatM$_{base}$ were used.

## 3.3   Simulations and analysis

CWatM was run from 1990–2019 and for future projections from 2020–2099 using the calibrated parameter sets. A total of 5 past simulations and 50 future simulations, using 5 GCMs and 2 SSPs, were performed for each CWatM$_{base}$ and CWatM$_{glacier}$ for the individual river basins. For global simulations, the same strategy was used. For the analysis, the period 1990–2019 and 2070–2099 were used because future changes were more pronounced at the end of the century.

Differences in past and future discharge between the baseline and the coupled model were evaluated close to the outlet of the river basins to assess the effect of explicit glacier inclusion on past and future discharge simulations using the ensemble mean values of the calibrations. In addition to projecting future discharge, climate impact studies often assess future changes in discharge by comparing future mean discharges to mean past discharges (e.g. Schewe et al., 2014; Gosling et al., 2017). This quantification of future change is essential for facilitating adaptation. Consequently, we also conducted comparisons between

the model setups to assess how the representation of glaciers influences our projections of future changes in discharge.

## 4   Results of selected river basins (5arcmin)

### 4.1   Past model evaluation

The magnitudes and seasonality of flow are generally well represented during the evaluation period (1990–1999) by the calibrated CWatM$_{base}$ and CWatM$_{glacier}$ (Fig. 3). Also for individual years, the discharge dynamics are well matched by the

simulations (Fig. S2.2).

The Fraser and Glomma basins have a similar discharge seasonality characterised by a melt regime with peak discharge during the early summer (May–July). Simulated discharge peaks later and declines earlier in the year than observed discharge for both basins and early winter discharge (Oct–Dec) is overestimated. The discharge regimes of the Rhone and Rhine river basins, on the other hand, are characterised by higher discharge during the winter months and relatively low discharge during

late summer (Aug–Sep) (Fig. 3). The simulations overestimate discharge for the Rhine river basin throughout most of the year, with a mean difference of 8%. In contrast, the simulations closely match observed discharge for the Rhone river basin. The distribution of flow magnitudes is well represented, but annual maximum discharges are underestimated for the Fraser basin and overestimated for the Rhine and Rhone river basins. The annual minimum discharge is underestimated for both model setups for all river basins except the Fraser (Fig. S2.4, S2.5).





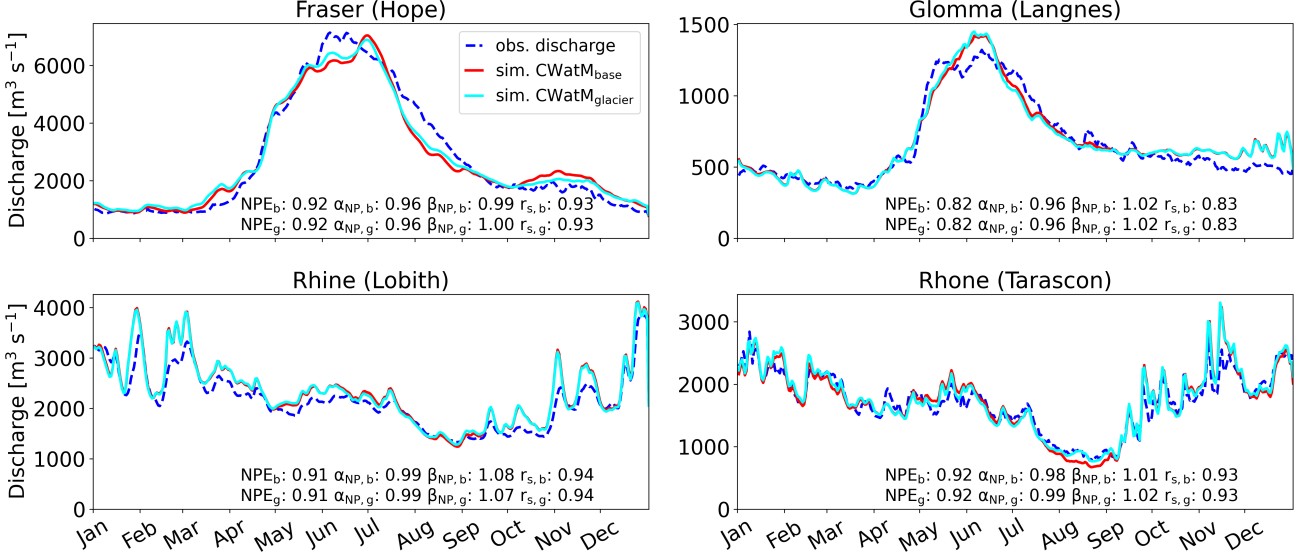

**Figure 3.** Mean hydrographs of the evaluation period (1990–1999) for the downstream station of the study basins. As insert the NPE and its error components over the whole evaluation period are given for CWaTM$_{base}$ (index b) and CWatM$_{glacier}$ (index g) ($\alpha_{NP}$: variability, $\beta_{NP}$: mean, $r_s$: dynamics; Pool et al., 2018). Values closer to 1 indicate a better match.

Observations are better matched by CWatM$_{glacier}$ for the Rhone river basin during August and to some extent for the Fraser river basin throughout most of the year. No noticeable differences between the model setups are evident for the Glomma and Rhine river basins . For the Rhone river basin, the differences between the two model setups are more pronounced at the upstream station (see Fig. S2.1). Simulations of CWatM$_{base}$ show a bias towards too low discharge in August for the Rhone river basin, which suggests that a relevant process during this time of the year is not represented well in this model setup. Using CWatM$_{glacier}$, summer discharge is well simulated (see Fig. 3, S2.1). This indicates that CWatM$_{glacier}$ can capture summer discharge better because glacier melt is explicitly included.

In general, discharge differences between CWatM$_{base}$ and CWatM$_{glacier}$ are marginal, which is also reflected by similar performance in the evaluation period. Model calibration can, to some extent, compensate for inadequate process representations within models (Duethmann et al., 2020; Knoben et al., 2020), and consequently, it can also mitigate the limitations arising from the absence of a glacier representation. During calibration of CWatM$_{base}$, a parameter set is chosen that exhibits a reasonable ability to simulate summer discharge by leveraging other processes within the model. For example, the calibrated parameters of CWatM$_{base}$ reduce evapotranspiration in the Rhone river basin to compensate for missing runoff from the glaciers in summer. The choice of parameters affects the partitioning of water to groundwater, soil, and the river and indirectly the length of the melt season. Therefore, model setups that do not include all relevant processes can perform fairly well but do not get the right answers for the right reasons (Kirchner, 2006). This presents a challenge when assessing whether an enhanced glacier





representation improves model simulations. Additionally, it becomes problematic in a changing environment, a topic discussed in the following section.

## 4.2 Future projections

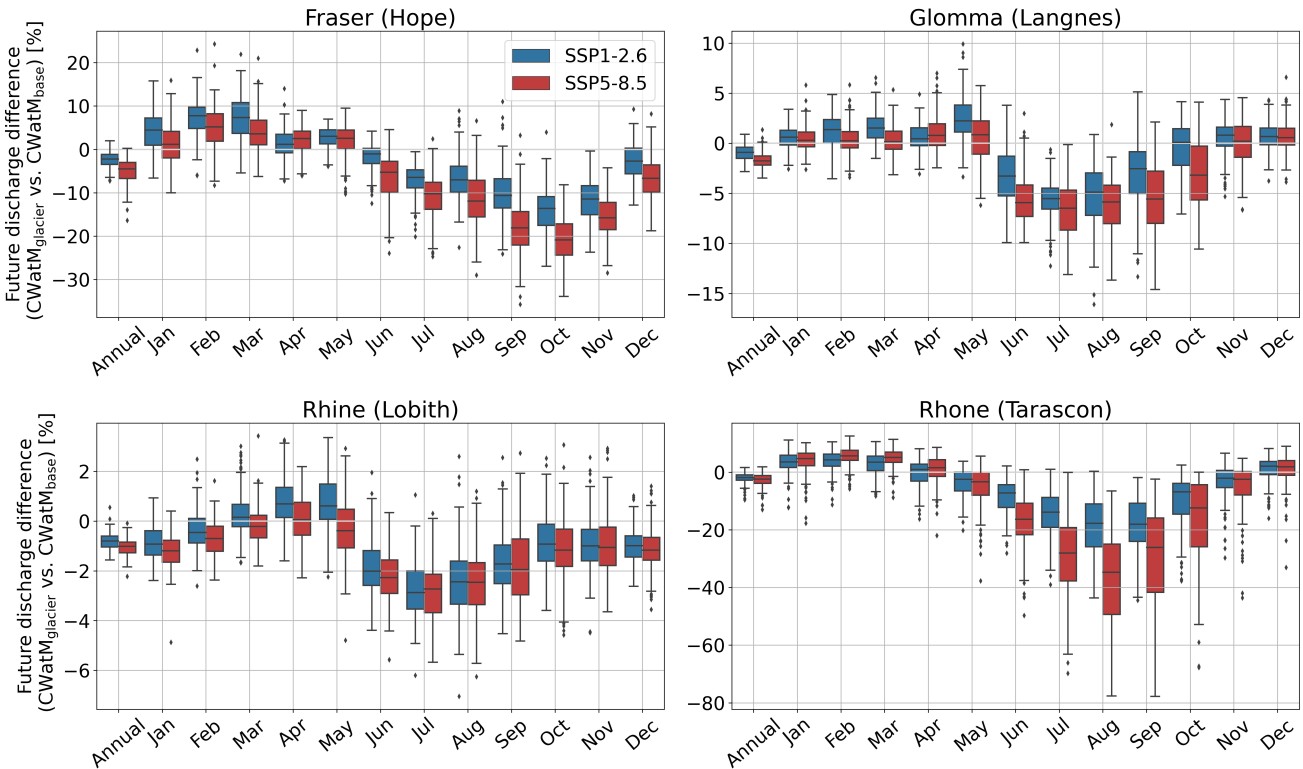

**Figure 4.** Relative difference in projected mean annual and mean monthly discharge of CWaTM$_{glacier}$ compared to CWatM$_{base}$ at the downstream gauges for 30 years from 2070–2099 using 5 GCMs (150 data points per boxplot). Differences are shown per SSP scenario which translate into global median warming levels of +1.9°C and + 4.2°C compared to pre-industrial time. Note that each subplot has a different y-axis scale.

The impact of enhancing the glacier representation in CWatM by coupling it with the glacier model OGGM becomes more
apparent at the outlets of our study basins when simulating future projections. Future simulated discharge of CWatM$_{glacier}$ is lower compared to CWatM$_{base}$ (Fig.4) with annual discharge differences of 1 to 4%. Monthly differences between the model setups are more pronounced and reach 20% for the Fraser and 35% for the Rhone but only 3% for the Rhine and 6% for the Glomma river basin (Fig. 4). The reason for more pronounced differences in the future compared to the past is that CWatM$_{glacier}$ captures glacier retreat and therefore changing glacier runoff during this century whereas CWatM$_{base}$ cannot capture this. The
calibration of CWatM$_{base}$ compensates for lacking glacier representation by adjusting parameters to fit the observed discharge in



the past. Therefore, it implicitly includes the glacier melt during the calibration period in the parameter sets. If these parameter sets are applied to future simulation periods with less glacier melt, future discharge is likely overestimated in the river basins by this model setup. The differences between the model setups are more pronounced for the higher emission scenario SSP5-8.5 and in months where glacier melt occurs (June to October). However, differences can also be seen during the snow melt period.

Comparing future discharge projections to past discharge simulations yields valuable information about the future change in discharge, which is an important metric for adaptation planning. Future change projections of discharge are more negative or less positive for CWatM$_\text{glacier}$ compared to CWatM$_\text{base}$ (Fig. 5), because CWatM$_\text{base}$ slightly underestimates discharge in the past during summer months and likely overestimates future summer discharge, which results in lower future changes.

    Differences between the model setups are largest for August where simulations with CWatM$_\text{glacier}$ project a change of -

29%/-58% for SSP1-2.6 / SSP5-8.5 for the Rhone river basin towards the end of the century, whereas the projected change with CWatM$_\text{base}$ is lower (-2%/-30%). On an annual scale, the differences are lower than for the summer months with +2%/-8% change in discharge for CWatM$_\text{glacier}$ and +6%/-6% changes for CWatM$_\text{base}$ (Fig. S3.2).

    For both model setups and all selected river basins, future change in discharge is positive in winter months and negative in summer months with a larger change for the higher emission scenario (Fig. 5). These future changes are consistent with

previous studies (Stahl et al., 2022). The range of projected change is large due to different regional projected changes in the forcing data of the GCMs.

    The effect of coupling is larger for the summer months, where glaciers contribute to discharge. Therefore, it is advisable to include a realistic glacier representation in the hydrological modelling of large river basins especially when looking beyond the mean annual cycle.

**4.3   Glacier melt contribution**

The previous section has revealed the effect of coupling a large-scale hydrological model to a glacier model and its added value in climate change studies for specific large river basins. This section exemplifies the possible use of the coupling to identify glacier melt contribution to discharge. The coupled model setup was run excluding glacier melt and the glacier areas, which are updated annually (CWatM$_\text{glacier,bare}$). This simulation was subtracted from CWatM$_\text{glacier}$ to derive the glacier contribution to

discharge including melt and rain on glaciers. Simulated annual glacier contributions to discharge show a drastic decline even under a moderate warming level of +1.9°C (SSP1-2.6 end of the century). It is projected that by the end of the century, these contributions will be less than 1.5% for the selected basins in Europe and North America (Fig. 6). In contrast, they were as high as 6% in the past. Consequently, glacier contributions to discharge are projected to become negligible on annual time scales in these river basins in the future. The monthly relative glacier contribution to discharge is largest in the Rhone river basin due to

the substantial percentage of basin area covered by glaciers. Additionally, discharge from non-glacierized areas is low during the months when glacier melt is occurring (see Fig. 3, S4.1). The simulated average glacier contribution in August diminishes from 37% to 14%/10% (340 m$^3$ s$^{-1}$ to 82 m$^3$ s$^{-1}$/38 m$^3$ s$^{-1}$) at the end of the century for a warming level of +1.9°C/+4.2°C (Fig. 6, S4.2). On an annual time scale, the glacier contribution to discharge is largest for the Fraser river because the percentage of basin area covered by glaciers is the largest.





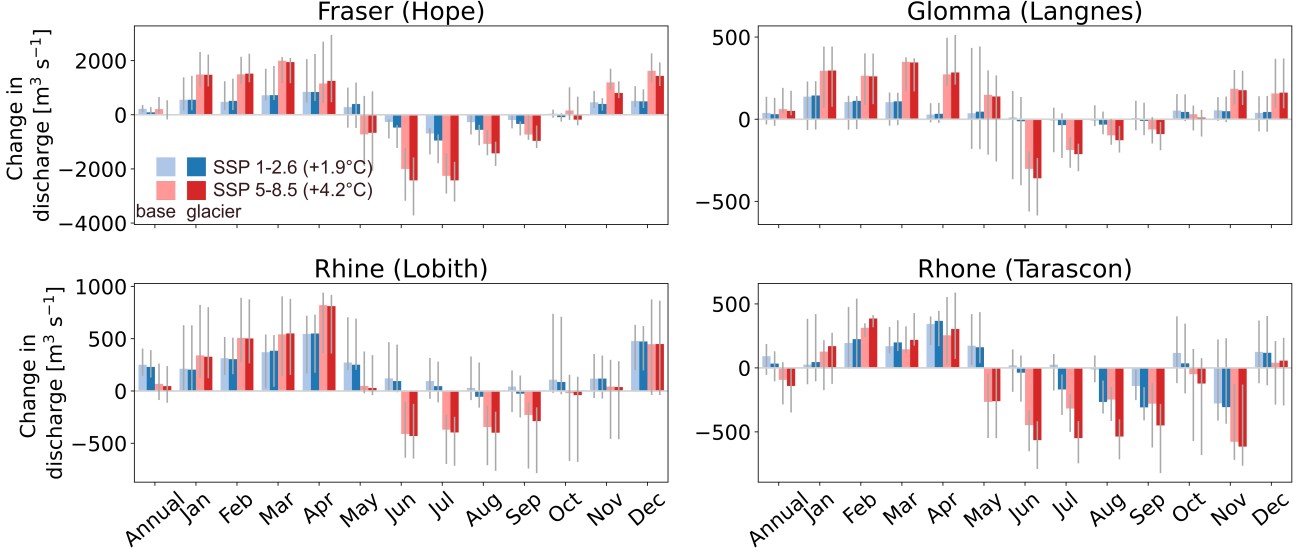

**Figure 5.** Absolute change in mean annual and mean monthly discharge at the downstream stations for the period 2070–2099 compared to 1990–2019 for CWatM$_{base}$ and CWatM$_{glacier}$, shown per SSP scenario which translates into global median warming levels of +1.9°C and + 4.2°C compared to pre-industrial time. The height of each bar indicates the median change based on the GCMs and the grey lines indicate the range between maximum and minimum change of individual GCMs.

## 5 Results of all glacierized river basins (30 arcmin)

### 5.1 Past model evaluation

We used six parameter sets within CWatM for the global evaluation to include parameter uncertainty in this study which is rarely done in large-scale hydrological modelling. The performance of these parameter sets for CWatM$_{base}$ was comparable to other large-scale hydrological models at 30 arcmin resolution (Müller Schmied et al., 2021; Sutanudjaja et al., 2018). Overall, simulations tend to overestimate observed discharge, which is in line with other studies (Müller Schmied et al., 2021; Wiersma et al., 2022) (see Fig. S2.6, S2.7).

Looking at glacierized river basins, observed discharge data are available for 30 out of 56 large-scale glacierized river basins worldwide (> 5000 km$^2$), mainly located in Europe and North America (Wiersma et al., 2022), with a glacier coverage ranging from 0.02% to 35%. The performance in simulating discharge increased or is at least similar for CWatM$_{glacier}$ compared to CWatM$_{base}$ for all these river basins (Fig. 7a), except for the Skagit river basin. This river basin is comparatively small (7850 km$^2$) and therefore likely not well represented at 30 arcmin resolution, where it covers only five grid cells.

The improvement in performance when explicitly including glaciers, as compared to CWatM$_{base}$, was largest for highly glacierized river basins (Fig. 7a). For example, the Copper river basin, which is 22% glacierized, exhibited substantial improvement in simulation performance when glaciers were explicitly included. In this case, the simulation of CWatM$_{glacier}$





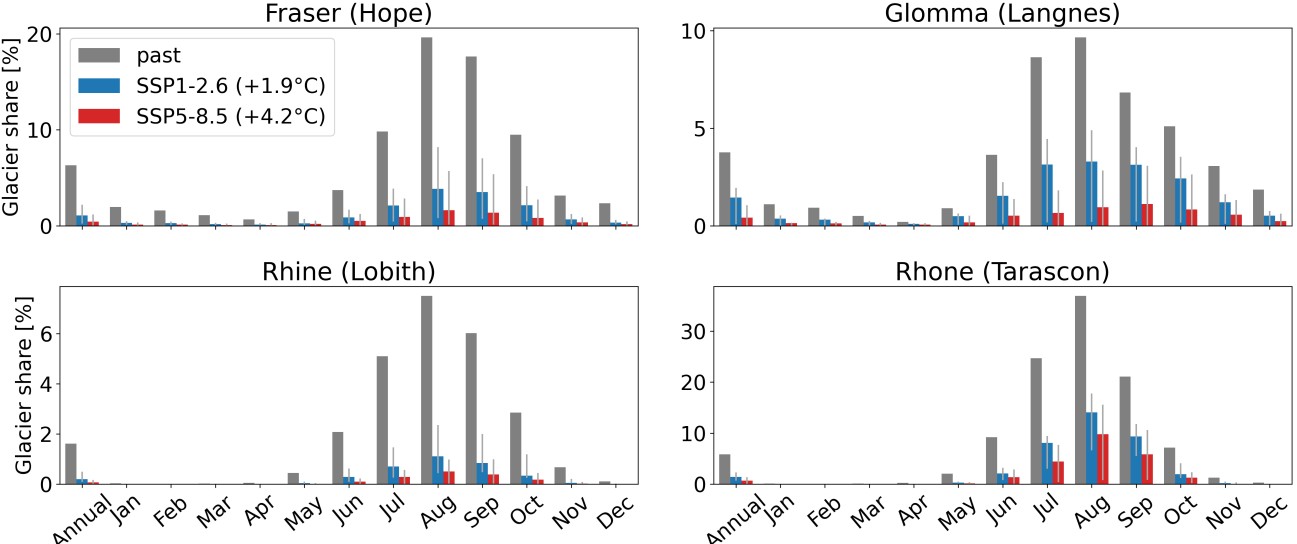

**Figure 6.** Relative mean glacier contribution to annual and monthly discharge (glacier share) at the downstream gauge for the period 1990–2019 and for the period 2070–2099 for two SSP scenarios which translate into global median warming levels of +1.9°C and +4.2°C compared to pre-industrial time. The height of the bar indicates median glacier share of GCMs and the grey lines indicate the minimum and maximum glacier share of GCMs.

matched the observations very well. The analysis of the runoff regime revealed that for CWatM$_{base}$, glacier runoff was missing as the relevant process in the Copper river basin during the summer months. Consequently, simulations with CWatM$_{base}$ failed to capture the observed behaviour (Fig. 7b).

In general, performance was very variable among the river basins for both model setups and all parameter sets. This is likely due to using global parameter sets which were not adjusted for each basin individually and inaccurate precipitation input. For example, the glacier inclusion improved the simulations of the Fraser river basin, but simulations were still unsatisfactory because discharge was highly underestimated (Fig. 7c). Thus, further effort in calibration is likely to enhance the advantages of CWatM$_{glacier}$ for global simulations. However, improving the global performance of large-scale hydrological models is a research field in itself and beyond the scope of this study.

### 5.2 Effect of coupling (past and future differences)

Differences between past simulations of CWatM$_{glacier}$ and CWatM$_{base}$ can be seen worldwide in large glacierized river basins (Fig. 8). The relative difference in monthly discharge between the two model setups can exceed 50% close to the glaciers and decreases further downstream. Nevertheless, differences between the model setups can also be detected downstream. For example, in September, the CWatM$_{glacier}$ simulations show 15% higher discharge in the Rhone basin, 26% higher discharge in the Glomma basin, and 17% higher discharge in the Amu Darya basin. This shows the influence of glaciers on discharge



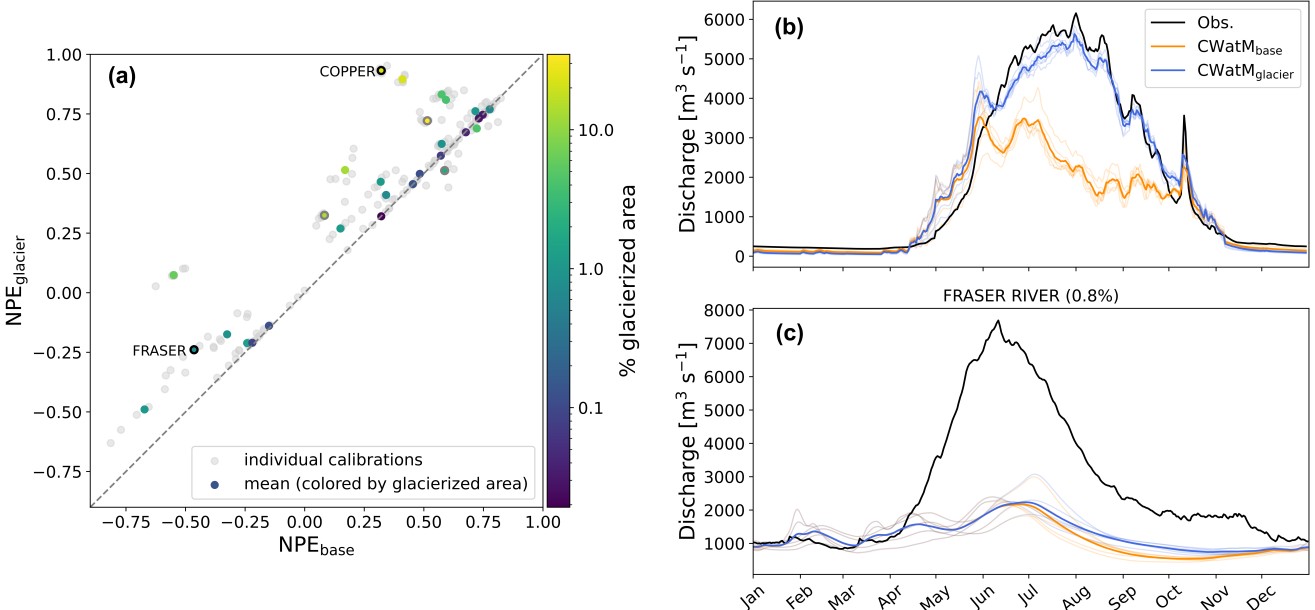

**Figure 7.** (a) Performance comparison using same discharge stations as presented in Wiersma et al. (2022) between CWatM$_{base}$ and CWatM$_{glacier}$ for individual calibrations (grey dots) and mean of all calibrations (coloured dots) for the 10 year period 2004 to 2013. The performance metric used is NPE (Pool et al., 2018). The Santa Cruz River basin lies outside the figure boundaries (NPE$_{glacier}$=-3.2, NPE$_{base}$=-195). Dots with grey outlines show basins smaller than 10,000 km$^2$. The median performance increase was 0.05 with an interquartile range from 0 to 0.2. Performance increased for 23 out of 30 basins. (b) and (c): Comparison of mean discharge of observations and simulations by CWatM$_{base}$ and CWatM$_{glacier}$. The individual lines show simulations of the different parameter sets and the bold line the mean of simulations.

in large river basins. The timing of the largest differences is delayed further downstream due to travel time through the river network. In winter, no glacier melt is expected; thus the change in model setup should not influence the winter discharge. However, a decrease in discharge is simulated for winter and spring months in CWatM$_{glacier}$ compared to CWatM$_{base}$ (see Fig. S5.2). The reason behind this is likely, the lack of glacier representation in CWatM$_{base}$ where other runoff processes occur on the glacier area instead of glacier runoff. Another explanation is the finer spatial resolution of elevations in OGGM (see

Table 1) and therefore a later snow melt occurrence in OGGM than in the same areas in CWatM.

On an annual average, the discharge simulated by CWatM$_{glacier}$ is larger or equal to that of CWatM$_{base}$. This is because the glaciers are diminishing, resulting in a negative change in storage that contributes to discharge. This additional water source simulated by OGGM is included in CWatM$_{glacier}$ but not in CWatM$_{base}$. The increase in annual discharge is larger for highly glacierized basins (Fig. S5.1). The annual difference between the model setups exceeds 10% for 15 basins and 5% for 22

basins.





**Figure 8.** Relative difference in average discharge for September between CWatM$_{glacier}$ and CWatM$_{base}$ during the period 1990–2019. Positive values indicate a larger discharge of CWatM$_{glacier}$. The outlines of large-scale glacierized river basins according to Huss and Hock (2018) (basin area > 5000 km$^2$, glacier coverage >0.01%) are shown in black. Note the difference in degree intervals.

Regarding future changes in river discharge, changes can be positive or negative depending on the regional changes in temperature and precipitation. In many river basins, the discharge in the month with the largest past glacier melt contributions is projected to decrease at the end of the century compared to the past (Fig. 9), whereas the annual discharge is projected to increase in many basins (Fig. S5.3). At the outlets of the 56 glacierized river basins, negative changes are exacerbated when glaciers are included explicitly in large-scale hydrological modelling, whereas positive changes are attenuated (Fig. 9). The reason is that simulated glacier melt is smaller at the end of the century than in the past and this negative change is included in CWatM$_{glacier}$. This pattern holds for the majority of river basins, with the exception of a few basins where simulated glacier runoff is greater or similar at the end of the century compared to the past because glacier peak water is reached later, most notably Santa Cruz, Tarim, Amu Darya and Jokulsa (Fig. S4.4). The difference in change between the model setups is more



pronounced for basins with a higher degree of glaciation. Overall, changes are more substantial for the higher emission scenario (Fig. S5.4).

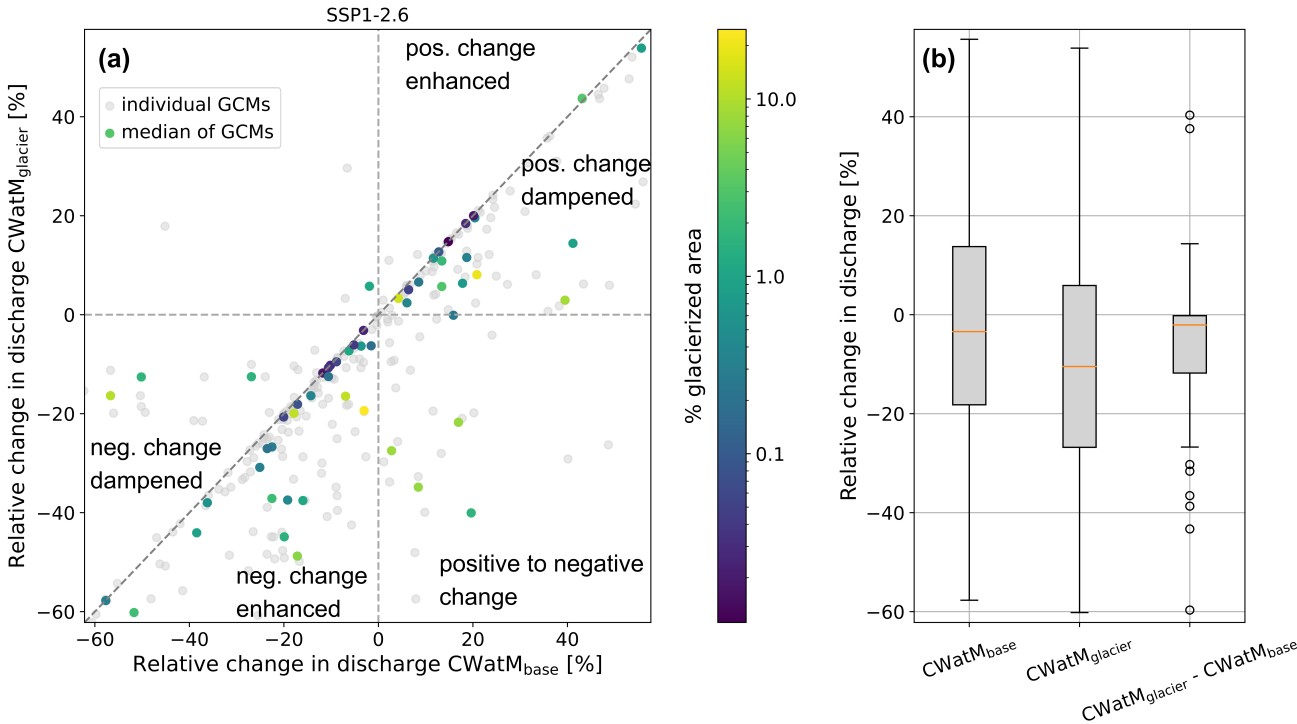

**Figure 9.** Comparison of relative future discharge change for the month with largest absolute glacier melt contribution in the past, at end of the 21$^{st}$ century, considering both CWatM$_{base}$ and CWatM$_{glacier}$, across 56 glacierized river basins for SSP1-2.6. (a) Colored dots show the median of all GCMs and grey dots show individual GCMs. (b) Boxplots show the relative future change of all basins for CWatM$_{base}$ and CWatM$_{glacier}$ and the difference between the two.

### 5.3 Glacier melt contribution

The simulated relative glacier melt contributions to discharge decreased for future periods at the outlets of all glacierized river basins, except for the Amu Darya and Tarim basins, both, on an annual average and during the month with the largest glacier melt contribution (Fig. 10). The decrease in glacier melt contribution to discharge is large in many basins. The highest monthly glacier contribution decreased by more than 10% in 22 river basins for SSP1-2.6 and 26 river basins for SSP5-8.5. It decreased by more than 20% in 12 river basins for SSP1-2.6 and 10 river basins for SSP5-8.5, respectively.

However, it should be noted that deriving glacier melt contributions can be highly uncertain if observed stream flow is not accurately represented in discharge simulations. An underestimation in discharge would result in an overestimation of relative





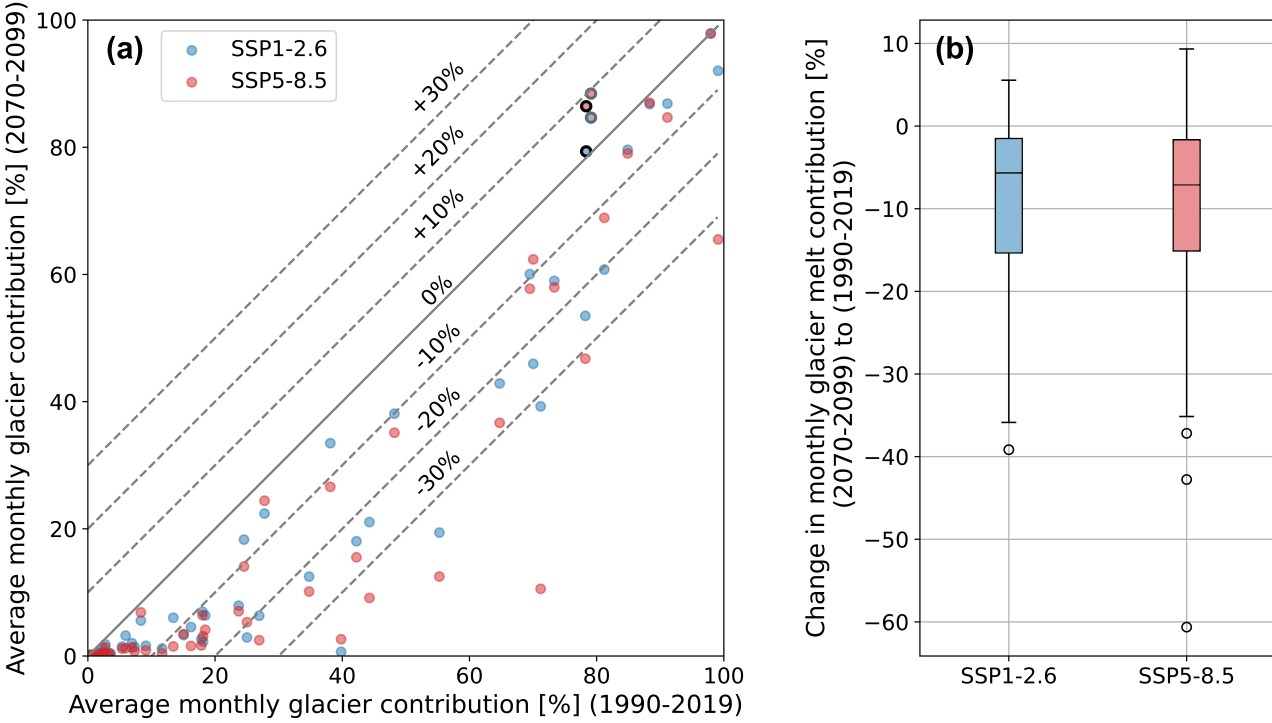

**Figure 10.** (a) Simulated glacier contribution to total basin discharge in the past (1990–2019) and end of the century (2070–2099) for the month with the highest relative glacier melt contribution to discharge in past. (b) The difference between the two periods, i.e. the simulated future change in glacier contribution. The outlined circles in (a) show the Amu Darya / Tarim basin (black/grey)

glacier melt contributions, whereas an overestimation of discharge would result in an underestimation of relative glacier melt contributions.

## 6 Discussion

### 6.1 Benefits of coupling

Glaciers were explicitly included in the simulations of the large-scale hydrological model CWatM, which simulates human

water use in addition to natural hydrological processes. Incorporating human water use is essential for studying large river basins. No less important is improving the representation of mountainous areas within large-scale hydrological models.

Therefore, including glacier runoff in large-scale hydrological modelling is an important step towards bridging the gap between assessments of natural environmental changes and their potential impacts on societies on larger scales. It is also a step towards better understanding the relevance of glaciers for water availability beyond small headwater catchments because the



coupling opens the possibility to estimate glacier contributions to discharge by running the model in different setups. Thus, the inclusion of glacier runoff offers the opportunity for further studies to delve into the impacts of glaciers on water availability.

To our best knowledge, our study is the first to explicitly incorporate glacier runoff generated on a daily resolution and accounting for ice dynamics into a large-scale hydrological model (cf. Cáceres et al., 2020; Wiersma et al., 2022). Our analysis demonstrated that explicitly including glaciers in both past and future simulations leads to notable changes in simulated discharge. Simulation differences were mainly present in the summer months, a period where water availability is especially relevant for irrigated agriculture. In specific years or months, the effect of including glaciers is likely larger than shown in this analysis, for which 30-year average values were used. This is potentially relevant for climate impact assessments with a focus on summer months or extreme years.

Climate change leads to an increase and subsequent decline in glacier melt volumes, occurring once the point of "glacier peak water" is surpassed (Huss and Hock, 2018). Therefore, the impact of improved glacier representation on future discharge changes depends on the period of comparison relative to glacier peak water. If the past period coincides with glacier peak water, the inclusion of glaciers in CWatM simulations results in a better representation of negative future discharge changes. This pattern holds for many river basins worldwide in this study when comparing 1990–2019 to 2070–2099 (cf. Fig. 9, S4.3). The rapid changes in glacier melt volumes throughout this century and the dependence of results on the selected period emphasise the relevance of explicitly including a glacier representation in large-scale hydrological modelling studies that focus on future discharge changes.

## 6.2 Caveats of coupling

The results of coupling a glacier model (OGGM) to a large-scale hydrological model (CWatM) are influenced by the limitations of each model. A major limitation in both models is uncertain precipitation data. Global glacier models such as OGGM are further limited by the scarcity of in-situ mass-balance observations available for model calibration (Schuster et al., 2023). The absence of globally available, temporally and spatially resolved mass-balance estimates pose a major challenge for the robust calibration of more enhanced mass-balance models (such as energy-balance models). Yet, such mass-balance models would be valuable to improve the representation of tropical glaciers, particularly since their current representation is limited by temperature-index models (Fernández and Mark, 2016). Moreover, model parameter choices affect glacier volume and runoff projections (Rounce et al., 2020; Schuster et al., 2023).

Large-scale hydrological models, including CWatM, are limited by the coarse representation of mountainous areas when typical resolutions of 10–50 km (5–30 arcmin) are used. We did not conduct a comparison between the simulations at different modelling scales for the selected river basins because simulation differences could not solely be attributed to scale differences but were also a result of different calibration procedures. At the global scale, the calibration and evaluation of models are limited by the availability of discharge data, which are biased towards mid-latitudes and temperate climates (Krabbenhoft et al., 2022). Additionally, large-scale hydrological models often lack a detailed calibration, resulting in poor model performance, especially in arid regions and regions with few discharge observations (Burek et al., 2020). Regionalized and calibrated parameter sets





that encompass all major hydrological processes would certainly improve simulations but are beyond the scope of this study. In the following section, the most important limitations of the models and their coupling are discussed in more detail.

**6.2.1 Precipitation correction**

In the hydrological model, a snow correction was only calibrated and applied when precipitation input was underestimated. This was the case for the Glomma and Fraser river basins, where the calibrated snow correction factor ranges from 1.1 to 2.6. In other cases, no snow correction factor was used to avoid the compensation of model flaws by correcting the input data. Thus, no snow correction factors were applied to the Rhine and Rhone river basins and the global simulations.

In global glacier models, inaccurate precipitation data at glacier scale is corrected using a precipitation factor (Hock et al., 2019). Different parameter combinations of precipitation factor ($p_f$), temperature bias and degree-day factor can represent the mass balance of a glacier equally well (Rounce et al., 2020). However, the precipitation factor has a strong impact on the simulated glacier runoff (Schuster et al., 2023). Here, it was chosen to best match the interannual variability of in-situ glacier mass balance data at the global scale ($p_f = 3$) and in the selected river basins ($p_f = 3$, for the Rhine $p_f = 2$). Thus,

precipitation correction is generally higher for the glacier model than for the hydrological model. One possible explanation is the higher elevation of glaciers where precipitation data are often less accurate. Another explanation is that the precipitation correction in OGGM accounts for processes that increase the snow input on glaciers such as avalanches or wind-blown snow but are not explicitly simulated (Rounce et al., 2020).

The difference in precipitation correction between OGGM and CWatM led to a larger precipitation input for CWatM$_{glacier}$

compared to CWatM$_{base}$. Across the 56 glacierized river basins, the mean difference was +5% for total precipitation and +17% for snowfall for the past period. Thus, differences between the model setups likely partly stem from the differences in precipitation correction. An extended analysis is presented in the supplements. CWatM$_{glacier}$ also outperforms CWatM$_{base}$ if precipitation input correction is equalized in both setups (Fig. S6.3).

We conducted additional global simulations with a precipitation factor of $p_f = 2$ to explore the importance of precipitation

correction in OGGM. Annual and August glacier melt volumes in the Rhine and Rhone River basins are around 25% lower for $p_f = 2$ compared to $p_f = 3$. The differences in total river discharge are much smaller when assessed based on the relative glacier melt contributions in the basins (see Fig. S6.4). For the highly glacierized Copper river basin, the discharge in July and August is around 20% lower using $p_f = 2$ instead of $p_f = 3$. This underscores the importance of conducting more comprehensive assessments of precipitation correction in future works to provide correct runoff information. A step towards

this is the work by Schuster et al. (2023) and a new option of variable precipitation correction for each glacier introduced in OGGM v1.6.0. However, this cannot compensate for missing precipitation data at high elevations (e.g. Viviroli et al., 2011; Shahgedanova et al., 2021).

**6.2.2 Glacier location in modelling grid**

The locations of glaciers within the river basin differ between real basin outlines and gridded basin outlines used for modelling.

Therefore, glacier melt can be incorrectly routed to a different basin. Although our routing approach for glacier melt and rain





on glaciers is consistent with the routing of precipitation in CWatM, it affects glacier melt contributions, especially at 30 arcmin resolution (Table S1). For example, in the Rhine river basin the average annual glacier melt volumes are 13%/6% lower on 30/5 arcmin resolution than for real basin outlines, for the Rhone they are 33%/5% lower. For simulations at 30 arcmin, attributing glacier melt to the correct river basin independent of grid cell resolution would enhance the glacier coupling.
However, such an implementation is not straightforward and has its own limitations, i.e. the change of the total basin area (Wiersma et al., 2022).

### 6.2.3   Differences in representation of mountainous terrain

Representation of elevation differences is distinct in OGGM and CWatM leading to differences in snow simulations (Table 1). These differences are inherent because of different spatial modelling scales (glacier scale vs. basin scale). Introducing sub-grid
elevation variability in large-scale hydrological models improves snow representation. However, the resolution, especially at 30 arcmin, is still too coarse to adequately capture small-scale elevation differences (Sutanudjaja et al., 2018). It's likely that glaciers are not precisely located at the elevation represented by the elevation zones in CWatM. Moreover, melt coefficients are not the same in the two models but calibrated to glacier (OGGM) and discharge (CwatM) characteristics, respectively. Thus, snow simulations are not identical in the models, leading to differences between CWatM$_{base}$ and CWatM$_{glacier}$ outside
the glacier melt season. It is difficult to investigate the effect of different snow representations on model results because the applied OGGM version does not differentiate between snow and ice, making it impossible to disentangle the snow and ice contributions. Nevertheless, due to the finer model resolution, the OGGM results are likely more realistic.

OGGM has no representation of hydrological processes through the subsurface. Therefore, it is only used to model the glacier-covered areas and not the emerging glacier-free areas. Runoff from glacier-covered areas is assumed to be routed to
surface runoff. Emerging glacier-free areas are simulated with CWatM, such that its runoff can infiltrate into the soil or bedrock because these hydrological processes are represented in CWatM. This distinction between glacier runoff and runoff from emerging glacier-free areas is essential, because runoff on ice-free areas may interact with the soil and bedrock. Therefore, our approach is an improvement compared to previous studies that do not apply such distinction (Huss and Hock, 2018; Wiersma et al., 2022).

### 6.2.4   Glacier melt contributions

In our approach, all glacier runoff was fed into surface runoff. We neglected the glacier melt contribution to groundwater recharge due to insufficient knowledge about its magnitude (Somers and McKenzie, 2020). This assumption likely overestimates glacier melt contribution to surface runoff. Glacier melt contributions to discharge were calculated by subtracting CWatM$_{glacier,bare}$ simulations, which excluded glacier areas from modelling, from CWatM$_{glacier}$ simulations, rather than subtract-
ing CWatM$_{base}$ from CWatM$_{glacier}$ simulations. This approach minimises the risk of incorrectly attributing model differences to glacier melt contributions (van Tiel et al., 2023). Nevertheless, differences may remain because CWatM$_{glacier,bare}$ excluded glacier areas from the highest elevation zone without explicitly considering the glacier elevation. This assumption does not represent the accurate elevation of each glacier, but the median accordance is high.





To understand the relevance of glaciers for water availability in large river basins it is important to derive the relative glacier
melt contribution to discharge. However, this ratio is highly dependent on the correct runoff regime representation, because
an underestimation (overestimation) of basin discharge leads to an overestimation (underestimation) of relative glacier melt
contributions. Thus, further improvements in the accuracy of large-scale hydrological models will help reduce the uncertainty
in relative glacier melt contributions.

## 6.3    Comparison to other studies

The performance of global hydrological simulations increases especially in highly glacierized river basins when glaciers are
explicitly included in modelling (Wiersma et al., 2022). Whereas Wiersma et al. (2022) demonstrated a decrease in model
performance in specific months for weakly glacierized basins when glaciers are explicitly included, our study did not find an
overall decrease in performance. This might be explained by the underestimation of discharge in many basins in the present
study, whereas discharge was mostly overestimated in Wiersma et al. (2022). Like Wiersma et al. (2022), we detected a decrease
in winter and spring discharge for the coupled model. Comparing glacier melt contributions to further studies is challenging
due to differences in modelling time periods and varying definitions of what constitutes glacier melt. Nevertheless, we compare
our results to other studies to see whether trends and magnitudes match.

The derived glacier contribution to runoff in the Rhine river basin at 5 arcmin resolution fits the results of a detailed study
on runoff components of the Rhine river basin (Stahl et al., 2016), e.g. 2.6%/4.2% at Lobith in August/September (1901–
2006) vs 7.5%/6% in this study (1990–2019). Differences between the studies can be explained by different periods and by a
different definition of glacier melt, which only comprises ice melt for Stahl et al. (2016) whereas it includes all runoff from
glacier areas in our study. Whereas our study suggests a glacier melt contribution of  40% for the Upper Indus previous studies
reveal a glacier contribution of  60% (Lutz et al., 2014) and  45% (Su et al., 2016). Nevertheless, all studies show a similar
pattern of high glacier contributions for the Upper Indus and rather low glacier contributions for the Upper Brahmaputra. The
glacier contributions we derived for the Himalayan basins are greater than the results reported by Kraaijenbrink et al. (2021).
Differences might stem from differences in evaporation estimates or other discharge components. Whereas Huss and Hock
(2018) show a strong decrease in relative glacier runoff for basins in Central Asia, our results show an increase in glacier
contribution to runoff in the Amu Darya and Tarim river basins. This projected increase in glacier contributions is consistent
with only a slight decrease in projected glacier volumes in these basins compared to all other regions in High Mountain Asia
at the end of the 21$^{st}$ century, as reported by Miles et al. (2021).

## 7    Conclusions

We added an explicit glacier representation in the large-scale hydrological model CWatM by developing a sequential coupling
with the global glacier model OGGM. The primary focus of evaluating this model coupling was its impact on future discharge
projections, given the anticipated substantial retreat of glaciers.



For simulations of individual large river basins, we found that calibration can compensate for the lack of glacier representation in the past. However, as a result of reduced glacier runoff, future discharge projections are smaller in the Fraser, Glomma, Rhine and Rhone basins with improved glacier representation. This leads to larger future discharge changes when glaciers are explicitly considered. Including glaciers globally without explicit calibration increases discharge in summer months, also at the outlet of large river basins. Performance improved in highly glacierized river basins and did not deteriorate in weakly glacierized basins. Glacier runoff is projected to be lower at the end of the 21$^{st}$ century compared to the recent past in most glacierized river basins worldwide. Therefore, positive future changes in discharge are attenuated and negative future changes are exacerbated when glacier representation is improved.

Thus, including glaciers in large-scale hydrological models is crucial for climate change impact assessments of water resources. We argue that an adequate glacier representation in hydrological simulations for large river basins is essential when studies focus on summer months or extreme years. Otherwise, climate change impacts on river discharge in glacierized river basins, even near the basin outlet, are likely to be underestimated. Such studies are relevant because seasonal and interannual variability potentially impact societies, the economy and the environment stronger than changes in the long-term annual averages. The coupled model setup can not only be used to improve the process representation in large river basins but can also be used to estimate glacier contribution to discharge. This is an important step in assessing the significance of glaciers for large river basins in both the past and the future. The collaboration between glacier and hydrological modelling communities should persist to further enhance the representation of glaciers in large-scale hydrological models.

*Code and data availability.* CWatM code is provided through a GitHub repository https://github.com/iiasa/CWatM (last access 13.10.2023) and the model version used for this study is provided via https://doi.org/10.5281/zenodo.10044318. Documentation and tutorials are available at https://cwatm.iiasa.ac.at/ (last access 13.10.2023). OGGM code is available through a GitHub repository https://github.com/OGGM/oggm (last access 13.10.2023) and the model version used for this study is available via https://zenodo.org/records/6408559. The documentation can be found at http://docs.oggm.org (last access 13.10.2023). The project website is used for dissemination http://oggm.org (last access 13.10.2023). The daily massbalance model used in this study is available through the GitHub repository https://github.com/OGGM/massbalance-sandbox (last access 13.10.2023) and the version used for this study is provided via https://doi.org/10.5281/zenodo.10055600. The pipeline to translate OGGM outputs to CWatM inputs is provided through a GitHub repository https://github.com/sarah-hanus/pipeline_oggm_cwatm and via Zenodo https://doi.org/10.5281/zenodo.10048089. The code for postprocessing simulation outputs, and creating the figures is publicly available via https://doi.org/10.5281/zenodo.10046823. The data set also contains parameter sets, (post-processed) simulation outputs and other relevant code and information used for this study. Climate data can be found on the ISIMIP server (Frieler et al., 2017). Global input data for CWatM at 5 arcmin and 30 arcmin is available upon request. Discharge data can be obtained from GRDC (2022) and HydroPortail (2022). Shapefiles of glacier outlines used in this study, namely Randolph Glacier Inventory v6.0, can be obtained from https://doi.org/10.7265/N5-RGI-60.



*Author contributions.* SH developed the model coupling, performed the simulations, did the analysis and drafted the manuscript. LS developed the daily mass balance model. LS and FM helped to configure OGGM for the model coupling and provided input on OGGM specifications and glacier modelling. PB developed the calibration procedure used for CWatM, helped to configure CWatM for this study and provided input on the hydrological model specifications. DV, FM and YW designed the study. All the authors discussed the results and

contributed to writing the final manuscript.

*Competing interests.* At least one of the (co-)authors is a member of the editorial board of Geoscientific Model Development.

*Acknowledgements.* The authors would like to thank Jan Seibert, Mikhail Smilovic and Ben Marzeion for useful discussions and support of this study. LS's contribution was funded by her DOC Fellowship of the Austrian Academy of Sciences at the Department of Atmospheric and Cryospheric Sciences, University of Innsbruck (No. 25928). LS's and FM's contributions were partly funded from the European Union's

Horizon 2020 research and innovation programme under grant agreement No. 101003687. This text reflects only the author's view and that the Agency is not responsible for any use that may be made of the information it contains.



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
