# Peer review of "Coupling a large-scale glacier and hydrological model (OGGM v1.5.3 and CWatM V1.08) – Towards an improved representation of mountain water resources in global assessments"

_EGUsphere, 2023_

## Author Comment (AC1)

**Author response to reviewer comments**

We thank Bettina Schaefli and Lander Van Tricht for their comments and constructive feedback for our manuscript. We provide our responses to their comments below. We used blue colour for our responses and *italic for passages from the manuscript*, where we highlighted in **bold implemented changes.**

Line numbers refer to the revised version of the manuscript.

**Author response to RC1**

*This paper on coupling of a large scale glacier model and hydrological model is very well written and presented; it represents an interesting contribution for large scale modelling as well as for other model coupling studies. It discusses in good detail the challenges related to such a loose one-way coupling (adapting the output of one model to be usable as input for the other model and dividing the model domain such that each model represents one domain). The actual results in terms of prediction future water resources for the selected basins are perhaps a bit less interesting because they are probably not unexpected and because there are still too many uncertainties regarding the actual water input and because of the crucial lack of details regarding the role of groundwater. These limitations could perhaps be a bit more prominent in the abstract. But this does not reduce the value of the paper.*

We highly appreciate the generally positive assessment of our manuscript and the helpful feedback by Bettina Schaefli. We added a sentence to the abstract to emphasize the uncertainties regarding actual water input. ***"The uncertainties in glacier-sourced runoff associated with inaccurate precipitation inputs require the continued attention and collaboration of the glacier and hydrological modelling communities."*** (l.15)

We acknowledge that large-scale hydrological models lack a detailed representation of groundwater due to the lack of hydrogeological information on global scale (de Graaf et al., 2017) and cannot provide the same detailed information as catchment-scale models. However, the role of large-scale hydrological models is to answer questions about water availability and water use globally, by comparing and contrasting different regions.

In the following we address the general and specific comments raised.

Detailed comments:

*1) The paper uses the terms runoff, surface runoff, discharge and streamflow, sometimes with unclear distinction of what is what; this mixing up of terminology is omnipresent in glacio-hydrological modelling and the actual meaning is often clear from the context but only for specialists; however, for gridded, large-scale models, it might be useful to specify if the model actually represents surface runoff (i.e. overland flow) and differentiate between simulated runoff and streamflow (I understand that the model has a transfer function to go from runoff to streamflow, see line 91, "routing);*

Thanks for this valuable comment. Surface runoff is defined as overland flow, and consists of direct runoff and interflow in the model. Runoff is the sum of surface runoff and baseflow. Streamflow is the runoff that is channelled and thus undergoes routing, as the referee correctly mentioned. Unfortunately, it is ambiguously named "discharge" in the model code. Discharge is the streamflow at a specific cross-section or point in space, e.g. at the discharge gauge.

We keep the term "glacier runoff" throughout the manuscript to describe water coming from glaciers, as this runoff is not yet channelled. We adapted the uses of the terms runoff, surface runoff, discharge and streamflow throughout the manuscript to match the definitions mentioned above.

We added a sentence to the manuscript to calrify the distinction between runoff, streamflow and discharge. ***"Streamflow is channelled and routed runoff, whereas discharge is the streamflow at a specific cross-section, e.g. at a discharge gauge."*** (l.35)

*2) Human water use is mentioned at several instances; would perhaps be important to discuss somewhere (if possible) the order of magnitude of the impact of human water on streamflow (in the considered catchments) compared to glacier -sourced streamflow*

In several studies, glaciers are mentioned to be of major relevance for human water use (Rounce et al. 2023, Huss and Hock, 2018). However, other studies disagree (Gascoin, 2024, Kraaijenbrink et al., 2021). Future studies with large-scale hydrological models that incorporate both human water use and glacier-sourced runoff can potentially provide a highly relevant contribution to this discussion as mentioned in l. 407-412.

Thus, we emphasized that large-scale hydrological models represent human water use as these models attempt to link water availability and water use on a global scale. This sets them apart from other models. CWatM considers domestic, industrial, livestock and irrigation water use (cf. Burek et. al, 2020). Our simulations included this human water use. However, contrasting and comparing human water use and glacier-sourced streamflow contributions is beyond the scope of this manuscript, even though it would indeed be an interesting future study. To disentangle the order of magnitude of human water use and glacier-sourced melt on streamflow, multiple simulations would be needed because glacier melt and human water use may interact: simulations including glacier-sourced melt and excluding water use, simulations including water use and excluding glacier-sourced melt and simulations including both factors.

We can however say that in glacierized river basins in Alaska and northern Canada, the impact of glaciers on streamflow will be considerably higher than the impact of human water use. In contrast, in river basins originating in High Mountain Asia, water is extensively used, therefore human water use impacts on streamflow are likely larger than the impact of glacier-sourced streamflow. For example, Döll et al. (2009) estimate the impact of water withdrawals on river discharges in their Figure 4d). They show that in many regions, reductions in streamflow are larger than 10% and in river basins originating from High Mountain Asia reductions can be larger than 40%, whereas in Alaska and Northern Canada, reductions by human water use are negligible.

We decided not to add more information on human water use in the main manuscript, because contrasting glacier-sourced melt with human water use was not the focus on this study and such an analysis would be a study by itself as indicated above.

*3) Perhaps it would be good to introduced a precise term for the water originating from glaciers, such as glacier-sourced rather than glacier melt? To avoid any mixing with "ice melt"?*

This is a very good idea. Thank you, we changed this in the new version of the manuscript and added a sentence to describe it, with reference to a new commentary published, which describes that "glacier melt" is used differently in different studies.

*"The definition of glacier melt varies among different studies (Gascoin, 2024). Here, we use the term "glacier-sourced melt" for the combination of ice and snow melt on glaciers to distinguish it from the sole ice melt."* (l.32-34).

*4) Perhaps it would be good to discuss somewhere what options the model has to "create" water if ice melt is missing (if we have negative mass balances, this represents a sources of water that the model is not able to simulate): either it drastically reduces ET or it requires a precipitation correction (more efficient)*

Yes, as the referee mentions, the model could "create" water by using a precipitation correction factor or by reducing ET. Per default, a snowfall correction factor is not applied in the model. In this study we only used it for the Glomma and the Fraser, as discussed in the answers to comment 13).

Calibrated parameters that affect the partitioning between ET and runoff are the crop factor, which adjusts the evapotranspiration of vegetation and the lake and river evaporation factor, a factor to adjust open water evaporation. Moreover, the soil depth factor adjusts the overall soil depth and therefore implicitly also affects evapotranspiration. Implicitly, also the preferential flow factor, which determines how much water is diverted to the groundwater reservoir, affects ET because water which is routed directly to the groundwater is not evaporated or transpired. In combination with the recession factor, which adjusts the contribution of baseflow to streamflow, the preferential bypass factor can reduce evapotranspiration.

We explained in lines 287-289, that for the Rhone river basin the parameter set changed such that ET was reduced for CWatM$_{base}$ compared to CWatM$_{glacier}$. For the other catchments, the difference in calibrated parameter sets between the model setups was less clear. For the upstream Rhone river basin, the crop factor, the preferential flow factor, the lake evaporation factor and the soil depth factors were lower for CWatM$_{base}$ than for CWatM$_{glacier}$, whereas the recession factor was higher, thus reducing ET.

This is shown in a figure added to the supplements (Fig. S2.1). It depicts the comparison of the calibrated parameter sets for CWatM$_{base}$ and CWatM$_{glacier}$ for the upstream gauge of the Rhone river basin, each boxplot comprises the five different parameter sets.

We rephrased our explanation in the revised manuscript:

*"During calibration of CWatM$_{base}$, a parameter set is chosen that exhibits a reasonable ability to simulate summer discharge by leveraging other processes within the model. Parameters influence the partitioning between evapotranspiration and runoff and the timing of runoff.* **Moreover, adjustments to the precipitation input (and thus the water volume) of the river basin can be made if a snowfall correction factor is applied.** *For example, the calibrated parameters of CWatM$_{base}$ reduce evapotranspiration and enhance groundwater contributions to streamflow in the Rhone river basin to compensate for missing runoff from the glaciers in summer (Fig. S2.1)."* (l.289-290)

*5) What do you recommend for modelling settings where there is no mass balance for model calibration, should the glacier be represented nevertheless or is the risk high that without calibration, the errors are too high to be useful for future projections? Or do your results not shed light on this?*

Direct glaciological mass balance measurements are only available for around 300 glaciers globally, which is around 0.1% of all glaciers (Maussion et al, 2019). Thanks to remote sensing

techniques, decadal mass balance data are now available for all glaciers worldwide (Hugonnet et al., 2021), but these data are lacking the year-to-year variations which are crucial for calibrating global glacier models. If we would only include glaciers with direct glaciological mass balance data in our hydrological models, we would omit most of the glacier volume, especially in Asia.

Whereas the uncertainty in glacier modelling due to missing mass balances may be substantial for individual glaciers, we argue that this uncertainty is reduced when modelling many glaciers as errors balance out. Thus, for small headwater catchments with only a handful of glaciers, errors may be too high to make useful projections if long-term glacier mass balance measurements are unavailable. However, for large river basins with many glaciers, the errors of individual glaciers become less important. Due to the strong change in available water from glaciers in future, we therefore still suggest including glaciers for future projections, at least when focusing on summer months and extreme years as mentioned in the manuscript.

We adapted the manuscript to clarify this:

*"Global glacier models such as OGGM are further limited by the scarcity of **long-term** in-situ mass-balance observations **(WGMS, 2020) and thus mostly rely on one observation per glacier (i.e., the 2000-2020 geodetic mass-balance average estimate of Hugonnet et al. (2021)). Nevertheless, we recommend including glacier runoff simulations for future discharge projections in large river basins with many glaciers, because glaciers are a rapidly changing water resource and influence future discharge changes, especially in the summer."** (l.430-434)*

The precipitation correction factor directly influences the interannual mass-balance variability. That means, glaciers with direct long-term mass-balance observations can be calibrated by tuning both the degree-day factor and the precipitation correction factor in order to match the average mass-balance and approximately the interannual mass-balance variability (Schuster et al., 2023). Thus, if we can constrain the precipitation correction better, which is very relevant for hydrological applications, we likely also match the interannual variability better even if we do not have yearly mass-balance observations available. Therefore, we emphasize better constraining the precipitation correction (compare to answer to comment 40 of reviewer 2).

**Even more detailed comment:**

*6) Is it mentioned somewhere if the selected catchments are past peak water or not? Was this a criterion?*

The selection of catchments was based on discharge data availability for calibration and evaluation of the model at the catchment outlet and a station upstream, closer to the glaciers. We set the criterion to 20 years of data availability between 1990 and 2019. This reduced the possible catchments to the Glomma, Rhine, Rhone, Danube and several catchments in North America. The final selection was based on the glacier coverage as a proxy for the glacier relevance and to ensure a geographical distribution. We therefore selected the Rhine, Rhone, Glomma and Fraser river basins as case studies.

Peak water was not a criterion for the selection. Peak water has most likely been passed in the Rhine and Rhone river basin before 2019 and is likely passed in the Glomma and Fraser river basin in these years (compare to Fig. S4.4 in the Supplements).

We added this to the discussion (l. 424-425): *"If the past period coincides with glacier peak water, the inclusion of glaciers in CWatM simulations results in a better representation of negative future*

*discharge changes. This pattern holds for many river basins worldwide in this study when comparing 1990--2019 to 2070--2099, **including the four selected river basins**"*

*7) Table 1: does the CWatM not use the DEM is input? Melt representation: seasonality of what, of the degree-day factor?*

CWatM uses the standard deviation of elevation per grid cell as input data. It is derived from the MERIT DEM (Yamazaki et al., 2019). The underlying assumption is that within a small spatial domain (each grid cell), elevation can be represented by a normal distribution. Therefore, elevation zones can be represented using the standard deviation of elevation in the grid cells. This allows us to have the same number of elevation zones in each grid cell, thus reducing the complexity of the calculations. Note that many global hydrological models do not even include a subgrid variability of snow (Telteu et al., 2021).

We changed the text in the Table: *"Meteo data (gridded), other gridded input maps, e.g. **elevation**, land use, soil and aquifer properties, drainage direction"*

The melt representation in CWatM includes the melt seasonality, where the snow melt rate varies depending on the day of year. Additionally, the melt rate is enhanced with increasing rainfall intensity (cf. Equation 1, Burek et al 2020). We changed the corresponding text to clarify this: *"Degree-day method **with melt rate seasonality** and the effect of increasing melt during rainfall, melt threshold at 0°C"*

*8) Figure 2: the Rhone catchment is missing? Can you add the names to the catchments?*

The Rhone catchment is located directly southwest of the Rhine catchment. To improve the Figure, we added names to the catchments and increased the visibility of the catchments' outlines.

*9) Can you say something about what the 12 calibrated parameters of CWaTM represent? What penalty function did you use for the snow cover error, what is excessive snow accumulation?*

The 12 calibration parameters represent adjustments for the major processes within the hydrological model (Table 2, Burek et al. 2020). The calibration parameter that interacts directly with the coupling is the degree-day factor and represents the snow melt rate. The other factors include a factor to adjust the evapotranspiration, the infiltration capacity, preferential flow, base flow and the runoff concentration among others. For a detailed explanation we refer to Burek et al. 2020, Table 2. We adapted the revised manuscript: *"A genetic algorithm (NSGA-II from python package DEAP, Fortin et al. (2012)) was used to calibrate 12 parameters in CWatM that govern major hydrological fluxes (Burek et al., 2020, **Table 2). These include factors to adjust the evapotranspiration, soil depth, preferential flow to groundwater, soil infiltration and groundwater contribution to streamflow."*** (l.230-231)

The penalty for the snow cover error is a simple function that ensures that the automatic calibration is not steered to parameter sets that represent well the discharge at the cost of snow cover representation, meaning that they accumulate snow to "get rid of water in the system". Therefore, we assume that for all the selected catchments the mean catchment snow cover should be close to 0m for at least two months per year. To build a penalty that is sensitive to snow accumulation, we set a threshold of 0.2m above which the objective function is negative:

$$O_{SC} = 1 - (S_{mean_{2months}}/0.2)$$

We then combine the objective function with the non-parametric KGE to obtain a single objective function for calibration.

$$O = 0.8 \cdot O_{KGE} + 0.2 \cdot O_{SC}$$

We acknowledge that our approach is rather simplistic, but we deem it sufficient for this study, as the focus is examining the effects of glacier inclusion in large river basins for global assessments rather than an in-depth calibration of specific basins. We added the description of the approach to the Supplements and adapted the main manuscript: *"It was combined with a penalty for the snow cover error to* **reduce multi-year snow accumulation (weight 0.8 / 0.2, cf. Supplements).**" (l.235-236)

*10) Perhaps seasonal flow duration curves would also be useful to show how well the model does?*

We exchanged the figure showing the flow duration curves in the Supplements for the figure below, which shows the overall flow duration curves and seasonal flow duration curves. We added a reference to it in the main manuscript. (l.276)

[Figure]

*11) Line 286: where do we see that the calibrated parameters reduce ET to compensate for missing runoff*

We added a figure of the parameter sets in the supplements and referred to it in the main text. (l.292) Compare with answers to comment 4). We also add a figure here showing the difference in mean ET over the simulation period and over the five parameter sets of $CWatM_{base}$ vs. $CWatM_{glacier}$. For the upstream part, which was calibrated against discharge at Lagnieu, the ET is considerably lower for $CWatM_{base}$. Only for pixels with glaciers, ET is higher for $CWatM_{base}$ because OGGM omits ET on glaciers. We also added a similar figure to the Supplements. (S2.2).

[Figure]

*12) Line 302: can you add a comment on why future discharge is likely overestimated*

Future discharge is likely overestimated in CWatM$_{base}$ if glacier-sourced melt is higher in the calibration period than in the future. We had explained this in l. 303 ff, but we added a sentence to clarify it.

*"The calibration of CWatM$_{base}$ compensates for lacking glacier representation by adjusting parameters to fit the observed discharge in the past. Therefore, it implicitly includes the glacier melt during the calibration period in the parameter sets. If these parameter sets are applied to future simulation periods with less glacier melt, future discharge is likely overestimated in the river basins **due to the stationarity of parameters**."* (l.306)

*13) Line 441: how do you know if precipitation is underestimated? Do you have some external estimate of ET ?*

For each catchment, we compared the long-term mean annual precipitation input and observed discharge to evaluate whether the precipitation input was reasonable. For the upstream station of the Fraser river, the mean annual discharge was 5% higher than the mean annual precipitation. Even if glaciers are an additional water input, we infer that the precipitation input is unreasonably low. For the upstream station of the Glomma river basin, the observed runoff coefficient (observed discharge / precipitation) was 0.93. We concluded that also for this river basin the precipitation was underestimated. Therefore, we included a snow correction factor in the calibration of the Fraser and Glomma. We clarified this approach in the revised manuscript:

*"In the hydrological model, a multiplicative time-independent snow correction was only calibrated and applied when precipitation input was underestimated, **based on comparing long-term mean precipitation and observed discharge.**"* (l.451)

*14) Thereafter: how is the precipitation factor applied, simple multiplicative factor on all time steps?*

Yes, the snow correction factor is a simple multiplicative factor. We clarified this in the manuscript:

*"In the hydrological model, a **multiplicative time-independent** snow correction was only calibrated and applied when precipitation input was underestimated."* (l.450)

*15) Line 484: differences outside the glacier melt season: I do not fully understand : s,now on glaciers is not modelled, not during nor after the glacier melt season, what is meant here?*

In OGGM, snowfall on glaciers is modelled and melt on glaciers is also modelled. However, there is no differentiation between snow and ice melt in the model output. As explained in the previous lines, the representation of snowfall and snowmelt in both models is not identical due to different melt coefficients (degree-day factors) and a different resolution of elevation which impacts snowfall and snow melt. In the results, we observe differences between the $CWatM_{base}$ and $CWatM_{glacier}$ also outside the ice melt season (July to September). We attribute these differences to differences in the representation of snowfall and snowmelt. However, we cannot confirm this attribution as OGGM does not differentiate between ice melt and snow melt in its output files.

We clarified this in the revised manuscript: *"Thus, snow simulations are not identical in the models, leading to differences between $CWatM_{base}$ and $CWatM_{glacier}$ outside the **ice melt season.**"* (l.495)

*16) It could perhaps be a bit more prominent how often the model domain is updated (glacier area), every year?*

Yes, the glacier area is updated annually because the glacier area in OGGM is updated each year. We had mentioned this in line 163. However, we agree that it is a good idea to reiterate this. Therefore, we added a sentence in the discussion: **"The glacier area was updated annually."** (l. 499).

*17) An automatic check of commas could further improve the readability of the paper*

Thank you for the suggestion. We thoroughly checked the manuscript again for commas.

**Author response to RC2**

*The paper described the integration of a large-scale open-source glacier model, OGGM, with a (large-scale) hydrological model (CWatM). Both the glacier and hydrology communities anticipated such direct coupling, its impacts, and its effects on statements about water availability. Consequently, the study by Hanus et al. is deemed a highly valuable contribution, being certainly worth publication in GMD.*

*The study couples two existing models, not directly, but by linking the hydrological model with the results of the glacier model, a one-way sequential coupling. These models were not drastically modified. Therefore, the novelty of the study lies more in discussing the results and the impact of incorporating glaciers as an additional water resource rather than in the development of a new model, although there are some new developments such as daily mass balance modelling, which I think is very interesting at itself as well.*

*The paper is well written, with clear descriptions of methods and formulations, making it easily accessible for others to apply and further develop the proposed method. Additionally, the main ideas are well presented. In summary, the study deserves publication, with some small technical revisions and clarifications which I mention below.*

We greatly value Lander van Tricht's positive assessment of our manuscript and thank him for the thorough revision and detailed comments to improve our manuscript, especially for the

improvements of the abstract. In the following, we provide answers to the specific comments raised by the reviewer.

***Specific comments:***

*1) Title: The last three words "in global assessments" could be removed. The focus of the paper is not on global-scale applications only. But I leave this to the authors to decide.*

We thank the referee for this comment. We would like to keep "in global assessments" as we think the novelty of the paper lies in the inclusion of glaciers in a global hydrological model, which is especially relevant for global assessments.

*2) Line 2: What do you mean by seasonality? Is this "runoff seasonality"?*

Yes, thank you for pointing this out. We changed it accordingly to *"… they are undergoing considerable changes in terms of area, volume, **magnitude and seasonality of runoff**."* (l.2)

*3) Line 5: Poorly or actually not even at all I think*

You are right, in most large-scale hydrological models, glaciers are not represented at all. However, some models have an extremely simplified glacier representation, such as CWatM prior to our study (cf. l.101-108). Moreover, some studies exist that included glaciers in other global hydrological models for specific purposes (Wiersma, et al. 2022, Caceres et al. 2020) and are mentioned in the Introduction (l.61-70). Therefore, we would like to keep the term "poorly represented".

*4) Line 6: "We evaluate" -> present tense?*

We use past tense throughout the manuscript for the description of what we did "We evaluated, we modelled, we coupled etc". We use present tense to describe established knowledge and to describe the results and implications of this work. We think this is a useful way to distinguish between tenses and would like to keep it in the revised version of the manuscript.

*5) Line 7: "Selected basins" is a bit vague. How many selected river basins? I think this can be stated here already. Later on I see the number if four? You can state "four" or mention their names.*

Good point, we added "four".

*6) Line 8: Which climate change scenarios? You could potentially add this information at this stage. For example, low and high emission climate change scenarios.*

Yes, we changed it to *"[…] focusing on **future** discharge projections **under low and high emission scenarios**."*

*7) Line 28: You could maybe give an example or two, for some of the basins that have passed peak water and some that are still heading towards the peak.*

We have extended the corresponding sentence as follows: *"Globally, glacier peak water has already been surpassed in around half of the glacierized river basins worldwide, **including the basins in Central Europe and western Canada. In most river basins originating in High Mountain Asia, such as the Amu Darya and Indus, glacier peak water is expected to occur around mid-century** (Huss and Hock, 2018)."* (l.30-33)

*8) Line 29-34: It's a bit unclear why you state these references here. Especially 29/31 focus on HMA but the study does not only.*

We wanted to mention studies that examine the relevance of glaciers for streamflow in large river basins worldwide or on a global scale, which is the focus of our paper. To clarify this, we added a sentence *"**Several studies examine the glacier melt contributions in large river basins**."* (l.35)

*9) Line 29 – 67: I have the feeling that this part could be written a bit more concisely. Some information is repeated and could be removed, making the introduction also a bit shorter.*

Thank you, we shortened the paragraphs.

*10) Line 71: Was -> is*

We would like to keep using past tense for what we did in our study. See answer to comment 4).

*11) Line 83: Could you mention here the number of selected basins?*

Yes, sure. We adapted the sentence: "*First, the framework is evaluated using* **four** *selected major river basins […]*"

*12) Line 112: Used -> used*

We would like to keep using past tense for what we did in our study. See answer to comment 4).

*13) Line 116-119: Past or present tense?*

See answer to comment 4).

*14) Line 142-145: Does this mean that precipitation is different at similar elevations in OGGM and CWatM?*

Yes, this is discussed in Section 6.2.1.

*14) Table 1: Does this mean that precipitation does not increase with elevation in OGGM?*

Yes, OGGM only includes an elevation-independent precipitation correction factor. That means, no precipitation lapse rate was applied in OGGM, as more observations are necessary to calibrate that parameter on a glacier-per-glacier level. Unlike other glacier models, where precipitation is assumed to increase with elevation, the OGGM developers do not believe that using a globally or regionally constant precipitation lapse rate adds value, because precipitation gradients are highly variable and likely not linear with elevation (e.g. Bookhagen and Burbank, 2006). We did not change the manuscript, as we already stated "No precipitation lapse rate" in Table 1.

*15) Line 177-183: Does this mean that you do not take into account the elevation of the glacierized areas? In other words, a large glacier which has its highest elevation in grid cell i loses the same fractional area as the grid cell j with a lower elevation? This simplification could be stated more clearly.*

Yes, this is correct. However, the elevation of the glacierized area is explicitly included in OGGM. Therefore, it is also implicitly considered in the glacier-sourced runoff that is used as input for CWatM. Yet, for determining the model domain in CWatM that has to be excluded to avoid double counting, we do not consider the explicit location where the glacier area is reduced (see Figure S1.1 in the supplements). We argue that this simplification is reasonable for applications in large river basins. The elevation zones in CWatM are a simplified representation of the elevation distribution in each grid cell (see the discussion Section 6.2.3) and therefore using a different fractional area reduction for each grid cell, depending on the elevation of the grid cell is not expected to enhance the modelling. We show in the manuscript that the largest impact of including the glacier-sourced runoff from OGGM comes from increased runoff in summer and changing glacier-sourced runoff in the future. Therefore, the decision of how much area of each glacier is reduced from which grid cell, if the respective glacier covers multiple grid cells, is minor, in our opinion.

To clarify this in the revised manuscript, first, we changed the order of the paragraphs, such that first the translation of glacier-sourced runoff to CWatM is explained and afterwards the translation of glacier area to CWatM input, to follow the order of relevance of the processes. Second, we added a sentence: "*However, the coupling was designed for large-scale modelling.*

*Therefore, these small-scale dynamics were **neglected when adapting the CWatM modelling domain**.*" (l.194)

*16) Line 208-209: Can be removed to make the section somewhat shorter and less repetitive*

Thank you, we removed the sentence.

*17) Line 210-214: These 5 lines can be written a bit shorter I have the feeling*

Thanks for pointing this out. We agree and have shortened it to "***The benefits and limitations of its applicability for both use cases are discussed.***" (l 213)

*18) Figure 2: Can you add the different basin names on this map? The distinction between Rhone and Rhine should also be clearer. Mention somewhere the glacier volume in the basins?*

Yes, we added basin names and made the outline of the basins clearer to differentiate between Rhine and Rhone. We also added the percentage of glacierized area and the glacier volume.

*19) Line 233-235: I am not familiar with these metrics. Can you add a bit more explanation on these metrics?*

Certainly. The KGE is a widely used metric in hydrology to assess the performance of hydrological models. It is a function that jointly assesses the error in the mean, the variability and the dynamic of discharge, giving each error type an equal weight. We added the following in the main text: *"As objective function, the non-parametric version of the Kling Gupta Efficiency (KGE) was used **which jointly evaluates the errors in the mean, the variability and the dynamics of the discharge** (NPE, Pool et al., 2018)."* (l. 234)

For an explanation of the snow cover error penalty, we refer to our answer to comment 9) of reviewer 1.

*20) Line 242-247: I do not understand the "five additional parameter sets". How does this differ from the global parameter set? Do you select different subsets of discharge data for calibration?*

We agree that the phrasing is not straightforward. We wanted to use the global parameter set that is also used in ISIMIP 3 simulations of CWatM to ensure consistency. We used the same calibration procedure that was used for obtaining this global parameter set, to obtain additional parameter sets. Repeating the calibration optimization procedure results in different parameter sets that are equally plausible because several parameter sets can perform equally well due to interactions between parameter sets.

We rephrased the sentence to ***"To consider parameter uncertainty we used six parameter sets obtained by calibration of CWatM_{base} including the parameter set used in ISIMIP 3 simulations of CWatM."*** (l.242 ff.)

*21) Line 248-249: Can you estimate the effect hereof? Would your results have been different when calibrating again? You show later in the paper that calibration is important (as it can partially compensate for the role of glaciers).*

For the global calibration, 601 discharge stations were used. The calibration procedure finds one parameter set that yields the maximum performance over all stations. This calibration procedure is different from the calibration at basin scale where calibration maximizes the performance at each discharge station. Of the 601 stations, 134 stations have some glaciers in their catchment area, out of which 63 stations have an upstream glacier area fraction >= 0.1%.

So, roughly only 10% of the stations used for calibration of the global model have a potentially relevant glacier influence. Therefore, we conclude that repeating the calibration for CWatM_{glacier} would only have minor influences on our results. This is due to the calibration procedure that maximizes performance over all stations, such that the weight of glacier-influenced discharge gauges is minimal. Note that global hydrological models often remain uncalibrated or poorly calibrated to date as mentioned in the same section.

We added a sentence to the manuscript to clarify this: *"**Repeating the global calibration for CWatM$_{glacier}$ would only have a minor influence on our results, because the weight of glacier-influenced discharge gauges in the global calibration is small.**"* (l. 250 f)

*22) Line 252: Why 50?*

This is just the result of having 5 parameter sets * 5 GCMs * 2 SSPs. We clarified this: "[…] 50 future simulations **(**using 5 GCMs and 2 SSPs**)** […]"

*23) Figure 3: Is the mean shown here? There seems to be a lot of variation. Insert -> inset?*

Yes, the Figure shows the daily mean hydrographs of the evaluation period (1990-1999), meaning that the daily discharge is averaged over this period, e.g. average 1$^{st}$ of January discharge over 30 years etc.. Such a variation over the year is expected. We clarified the caption: *"**Daily mean hydrographs**"*

Thanks, we changed "insert" to "inset".

*24) Line 277: pronounced -> pronounced*

Yes, changed.

*25) Line 277: remove the space after "basins".*

Yes, changed.

*26) Line 279: Very cool to see!*

Thank you.

*27) Line 283-285: I'm wondering which process than matter in summer to provide discharge if glaciers are not represented?*

The calibration tries to tune the model to mimic the glacier runoff in summer, if the hydrological model is calibrated without considering glaciers and glacier runoff is a relevant contribution to discharge. Thus, evapotranspiration is reduced, such that less water is "lost" to the atmosphere. Also, the preferential flow of water to groundwater and the contribution of groundwater to streamflow (called baseflow) are enhanced. Therefore, the model is tuned towards increased contributions from precipitation (due to lower evapotranspiration) and groundwater to streamflow.

We added a corresponding figure to the supplements. Please also look at the answers to comment 4) and 11) of reviewer 1, which address similar comments.

To clarify it in the manuscript we added: "*For example, the calibrated parameters of CWatM$_{base}$ reduce evapotranspiration **and enhance groundwater contributions to streamflow** in the Rhone river basin to compensate for missing runoff from the glaciers in summer (Fig. S2.1).*" (l.291)

*28) Line 315: Studies -> Could you add more references here?*

Yes, we added two more studies, namely Rottler et al. (2020) and Schneider et al. (2013).

*29) Line 317: Contribute -> Contribute most?*

Yes, we changed it.

*30) Line 328-329: Repetition. In the previous sentence -> 1.5% is negligible. This can be removed (for me).*

Yes, we deleted the sentence and added to the previous "By the end of the century, these contributions are projected to be less than 1.5% for the selected basins in Europe and North America **and thus negligible**."

*31) Line 325-334: This section could be rewritten somewhat less extensively. For example,*

Thanks, we combined some sentences to make it more concise.

*32) Line 344: The performance increased -> Improved?*

Yes, thanks.

*33) Line 350: Idea to show or more mention which processes compensate for the missing glaciers?*

For the global simulations CWatM$_{base}$ does not compensate for missing glaciers, because the global calibration of CWatM is simplistic and glaciers only play a minor role (compare to answers to comments 21) and 27)). Therefore, we did not change the manuscript in response to this comment.

*34) Line 350: Idea to show how the importance or improvement evolves as a function of glacier fraction?*

Thanks for this suggestion. We show a figure below that plots the glacier area fraction against the difference in performance. We did not include the figure in the main text as Figure 7a) implicitly includes the performance improvement in relation to the glacierized area. However, we think it is a good idea to add a figure that focuses on it explicitly in the supplements. (Figure S5.2)

[Figure]

*35) Line 379-386: What about the precipitation projections?*

Future discharge change depends on future changes in precipitation and temperature as mentioned in l. 379. Precipitation projections are the same for CWatM$_{base}$ and CWatM$_{glacier}$ simulations. Thus, changes in the precipitation projections affect the future discharge change. However, they are not expected to change the results regarding the effect of glacier inclusion mentioned in l. 382 ff.

We changed the manuscript to clarify this: *"The reason is that simulated glacier-sourced melt is smaller at the end of the century than in the past and this negative change is included in*

*CWatMglacier, whereas precipitation projections are identical for both model setups."* (l. 385)

*36) Line 388-390: Can you say something about why in these basins there is no net decrease? Is glacier runoff by the end of the century still larger than at present due to the large glaciers in these basins still reacting to climate change?*

For the Amu Darya and Tarim river basin, the glacier-sourced melt in 2070-2099 is similar or larger than in 1990-2019. This is also the case for a few other river basins, namely the Copper, Indus, Jokulsa a Fjollum, the Santa Cruz and for SSP5-8.5 also for the Yukon (cf. Fig. S4.4). However, these other river basins experience an increase in future discharge, except for the Santa Cruz river basin (which is not well represented by the simulations, cf. Fig. 7 and S6.4). Thus, the relative glacier-sourced melt contribution decreased for these river basins. For the Santa Cruz river, they remain similar. The Amu Darya and Tarim river basins experience a decrease in future discharge, thus the relative glacier-sourced melt contributions increase.

We added a sentence to the revised manuscript to clarify this: ***"In the Amu Darya and Tarim basins, the projected future glacier-sourced melt is either larger or similar to the past. This, combined with projected decreased discharge, increases relative glacier-sourced melt contributions."*** (l.394 f)

*37) Line 401: Which processes are considered regarding human water use? Can you give some more details about this as it appears to be very important*

CWatM considers domestic, industrial, livestock and irrigation water use (cf. Burek et. al, 2020). We clarified this in the revised manuscript: "*Additionally, it can **simulate irrigation, industrial, domestic and livestock** human water use and reservoir regulations.*" (l.94)

A longer answer about human water use is given to comment 2) of reviewer 1.

*38) Line 411-413: Interesting statement. Using your results, could it be possible to estimate the maximum/minimum contribution of glaciers? This could be something interesting to mention already earlier in the paper*

We refrain from analyzing individual years in our study, as this is beyond the scope of this work. However, using our results for the four selected basins, glacier contributions of individual months in specific years can be assessed and thus a minimum and maximum monthly glacier contribution can be estimated. For the global assessment, the calibration is probably too simplistic to robustly estimate glacier contributions of months in individual years and thus assess the minimum and maximum glacier contributions. We added a sentence to Section 4.3: ***"Note that we only show 30-year averaged results here (Fig. 6), but a similar approach could be taken to estimate glacier-sourced melt contributions in individual months and years."***

*39) Line 454-458: This is actually quite important and should be mentioned earlier in the manuscript*

We added a sentence to Section 2.3 to clarify this early on: "*In other cases **and for global simulations,** no snowfall correction was applied. **This leads to larger precipitation inputs in the coupled model compared to the baseline model, a topic further discussed in Section 6.2.1.**" (l.148 f)

*40) Line 460-466; The differences are large. I think this should be clearer in the manuscript while also putting more forward the importance of better constraining the precipitation factor.*

We agree that differences are considerable. This was also the outcome of previous studies mentioned in the same section (Schuster et al., 2023, Rounce et a., 2020).

We adapted the last sentence of the conclusion to emphasize this: *"**Yet, considerable uncertainties in glacier-sourced runoff remain, which need to be addressed by better constraining the precipitation input and correction in global glacier modelling. Thus,** the*

*collaboration between glacier and hydrological modelling communities should persist to further enhance the representation of glaciers in large-scale hydrological models.”* (l.562 f)

And added a sentence to the abstract: ***“The uncertainties in glacier-sourced runoff associated with inaccurate precipitation inputs require the continued attention and collaboration of the glacier and hydrological modelling communities.”*** (l.15 f)

*41) Line 490: What about the glacier runoff routing? Is glacier runoff instantaneously supplied to CWatM or can there be a delay depending on glacier size? Is refreezing included in OGGM? If not, this could also be stated in this paragraph.*

Yes, glacier runoff is supplied to CWatM instantaneously. This is also mentioned in section 2.4.3. There is no delay depending on the glacier size. Refreezing is not included in OGGM. We added a corresponding sentence in Section 2.2: "***The model version does not consider refreezing.”*** (l.121)

*42) Line 515: Further -> Other?*

Yes, thanks.

**References**

Bookhagen, B., & Burbank, D. W. (2006). Topography, relief, and TRMM-derived rainfall variations along the Himalaya. Geophysical Research Letters, 33(8).

Burek, P., Satoh, Y., Kahil, T., Tang, T., Greve, P., Smilovic, M., Guillaumot, L., Zhao, F., and Wada, Y.: Development of the Community Water Model (CWatM v1. 04)–a high-resolution hydrological model for global and regional assessment of integrated water resources management, Geosci. Model.Dev., 13, 3267–3298, 2020.

Caceres, D., Marzeion, B., Malles, J. H., Gutknecht, B. D., Muller Schmied, H., and Doll, P.: Assessing global water mass transfers from continents to oceans over the period 1948–2016, Hydrol. Earth Syst. Sci, 24, 4831–4851, 2020.

de Graaf, I. E., van Beek, R. L., Gleeson, T., Moosdorf, N., Schmitz, O., Sutanudjaja, E. H., & Bierkens, M. F. (2017). A global-scale two-layer transient groundwater model: Development and application to groundwater depletion. Advances in water Resources, 102, 53-67.

Döll, P., Fiedler, K., & Zhang, J. (2009). Global-scale analysis of river flow alterations due to water withdrawals and reservoirs. Hydrology and Earth System Sciences, 13(12), 2413-2432.

Gascoin, S.: A call for an accurate presentation of glaciers as water resources, Wiley Interdisciplinary Reviews: Water, 11, 2024

Hock, R., Bliss, A., Marzeion, B., Giesen, R. H., Hirabayashi, Y., Huss, M., Radi´c, V., and Slangen, A. B.: GlacierMIP–A model intercomparison of global-scale glacier mass-balance models and projections, J. Glaciol., 65, 453–467, 2019.

Hugonnet, R., McNabb, R., Berthier, E., Menounos, B., Nuth, C., Girod, L., Farinotti, D., Huss, M., Dussaillant, I., Brun, F., et al.: Accelerated global glacier mass loss in the early twenty-first century, Nature, 592, 726–731, 2021.

Huss, M. and Hock, R.: Global-scale hydrological response to future glacier mass loss, Nat. Clim. Change, 8, 135–140, 2018.

Kraaijenbrink, P. D., Stigter, E. E., Yao, T., and Immerzeel, W. W.: Climate change decisive for Asia's snow meltwater supply, Nat. Clim. Change, 11, 591–597, 2021.

Maussion, F., Butenko, A., Champollion, N., Dusch, M., 670 Eis, J., Fourteau, K., Gregor, P., Jarosch, A. H., Landmann, J., Oesterle, F., et al.:

The open global glacier model (OGGM) v1. 1, Geosci. Model.Dev., 12, 909–931, 2019.

Rottler, E., Francke, T., Bürger, G., and Bronstert, A.: Long-term changes in central European river discharge for 1869–2016: impact of changing snow covers, reservoir constructions and an intensified hydrological cycle, Hydrol. Earth Syst. Sci, 24, 1721–1740, 2020.

Rounce, D. R., Khurana, T., Short, M. B., Hock, R., Shean, D. E., and Brinkerhoff, D. J.: Quantifying parameter uncertainty in a large-scale glacier evolution model using Bayesian inference: application to High Mountain Asia, J. Glaciol., 66, 175–187, 2020.

Rounce, D. R., Hock, R., Maussion, F., Hugonnet, R., Kochtitzky, W., Huss, M., Berthier, E., Brinkerhoff, D., Compagno, L., Copland, L., et al.: Global glacier change in the 21st century: Every increase in temperature matters, Science, 379, 78–83, 2023.

Schneider, C., Laizé, C., Acreman, M., and Flörke, M.: How will climate change modify river flow regimes in Europe?, Hydrol. Earth Syst. Sci, 17, 325–339, 2013.

Schuster, L., Rounce, D. R., and Maussion, F.: Glacier projections sensitivity to temperature-index model choices and calibration strategies, Ann. Glaciol., p. 1–16, https://doi.org/10.1017/aog.2023.57, 2023.

Telteu, C.-E., Muller Schmied, H., Thiery, W., Leng, G., Burek, P., Liu, X., Boulange, J. E. S., Andersen, L. S., Grillakis, M., Gosling, S. N., Satoh, Y., Rakovec, O., Stacke, T., Chang, J.,Wanders, N., Shah, H. L., Trautmann, T., Mao, G., Hanasaki, N., Koutroulis, A., Pokhrel, Y., Samaniego, L.,Wada, Y., Mishra, V., Liu, J., Doll, P., Zhao, F., Gadeke, A., Rabin, S. S., and Herz, F.: Understanding each other's models: an introduction and a standard representation of 16 global water models to support intercomparison, improvement, and communication, Geosci. Model.Dev., 14, 3843–3878, 2021.

Wiersma, P., Aerts, J., Zekollari, H., Hrachowitz, M., Drost, N., Huss, M., Sutanudjaja, E. H., and Hut, R.: Coupling a global glacier model to a global hydrological model prevents underestimation of glacier runoff, Hydrol. Earth Syst. Sci, 26, 5971–5986, 2022.

Yamazaki, D., Ikeshima, D., Sosa, J., Bates, P. D., Allen, G. H., and Pavelsky, T. M.: MERIT Hydro: A high-resolution global hydrography map based on latest topography dataset, Water Resour. Res., 55, 5053–5073, 2019.